# FACTOR: Factoring Complexity and Context Length in Long-Context Model Evaluation

## ABSTRACT

Large language models (LLMs) with extended context windows are gaining attention. However, whether these resource-intensive LLMs indeed surpass simpler Retrieval Augmented Generation (RAG) techniques remains debatable. We precisely delineate differences between long-context LLMs and RAG methods, emphasizing the long-context reasoning abilities of LLMs. Existing benchmarks for long-context models often focus on information retrieval, hindering the assessment of reasoning over extended contexts. We introduce the **FACTOR** benchmark (**F**actoring **A**nalysis of **C**omplexity and **T**extual **C**ontext in **R**easoning). FACTOR consists of two suites of tasks, covering both the *symbolic* and *real-world* facets of reasoning evaluation. Both suites are carefully created to delineate task complexity and context length when evaluating LLMs. We present detailed evaluations of popular LLMs on FACTOR. Besides accuracy scores, we also model the relationship between accuracy and task complexity. A simple but consistent log-linear relationship works surprisingly well across various models. From the log-linear relationship, two explainable parameters, the slope or Complexity Decay Factor (CDF) and the y-intercept or Contextual Decay Offset (CDO), are shown to offer separate and insightful measures of the models' complex reasoning and long context innate ability. Our findings highlight distinct failure modes linked to task complexity and context length, underscoring the unique reasoning capabilities of long-context LLMs unattainable by RAG methods.[1]

## 1 INTRODUCTION

Recently, large language models (LLMs) with extended context windows have gained attention in real-world applications (Achiam et al., 2023; Team et al., 2024). With the advent of next-generation models (Dubey et al., 2024), context lengths of up to 128K tokens are becoming the new norm. Despite these advancements, a persistent debate exists (Li et al., 2024; Yu et al., 2024) regarding whether sophisticated and resource-intensive long-context LLMs genuinely offer advantages over more straightforward and cost-effective Retrieval Augmented Generation (RAG) techniques. This paper aims to precisely delineate the differences between what LLMs can accomplish with extended context capabilities and what is attainable through RAG methods. We contend that long-context LLMs possess unique long-context reasoning abilities that are inherently challenging for RAG-based methods to replicate.

While existing benchmarks, such as RULER (Hsieh et al., 2024) and ∞Bench (Zhang et al., 2024), cover a range of tasks and are becoming the new paradigm for evaluating the long-context models, they often fail to capture the fundamental distinctions between the reasoning capabilities of long-context LLMs and the retrieval strengths of RAG methods. Specifically, existing benchmarks exhibit two key limitations that inadvertently favor RAG-based approaches: (1) they heavily present tasks focused on tasks that assess the model's retrieval ability for long context (key characteristics being the complexity independent of context length) and (2) even though some tasks indeed see complexity increases with context length, e.g. Variable Tracking (Hsieh et al., 2024), they are often still too simple, and doesn't require model's reasoning ability to solve. We empirically showed that simple RAG techniques easily get perfect scores on the Variable Tracking tasks. Using the MPnetv2 Song et al. (2020) as the encoder and Llama3.1 8B Instruct Dubey et al. (2024) model as the generator, the system achieves 100% and 98% accuracy on 131k and 1M context length with only 1024 actual context during the retrieval. Experiments show that solving VT is viable through simple backtracking only the occurrence of the query variable without grasping the entire problem setting and states of other variables. We contend that benchmarks hide the unique but impressive reasoning advantages that

[1]Revision during Rebuttal are marked in color blue.

---

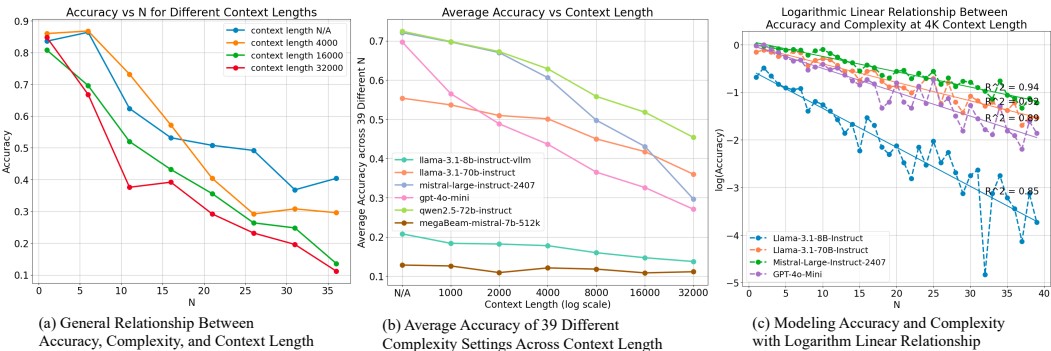

(a) General Relationship Between Accuracy, Complexity, and Context Length

(b) Average Accuracy of 39 Different Complexity Settings Across Context Length

(c) Modeling Accuracy and Complexity with Logarithm Linear Relationship

Figure 1: (a) presents representative performance of Llama-3.1-70B-Instruct on FACTOR. From the trend, we can see that as N or the complexity increases, the accuracy decreases. Also, as the context length of the question prompt increases, the curve shifts downward. (b) shows the ranking of mainstream long-context LLMs on the average accuracy taken from all 39 different complexity settings N. (c) presents our attempt to model accuracy versus complexity for various models generally. Surprisingly, we observe that the logarithm of FACTOR accuracy linearly correlates nicely with the complexity N.

long-context LLMs can provide over RAG techniques. Therefore, the long-context language model needs a benchmark that can evaluate the LLM ability beyond retrieval. More detailed evaluation in Section 3.

Therefore, in this paper, we introduce the **FACTOR** (**F**actoring **A**nalysis of **C**omplexity and **T**extual **CO**ntext in **R**easoning) benchmark, designed to systematically evaluate language models' long context complex reasoning ability through carefully curated synthetic tasks that require models' grasp of the entire problem to succeed. Similar to previous benchmarks, FACTOR disentangles task complexity and context length, each can be independently varied. Specifically, the FACTOR benchmark relies on a suite of synthetic task generators by adjusting the following two key knobs for independently controlling task complexity and context length.

- **Number of Variables (Task Complexity)**: Defines the complexity of the reasoning task via the number of interdependent variables. For most evaluations, the number of variables is limited to less than 40.

- **Length of Filler Text (Context Length)**: To independently control the task complexity, we insert the question prompt with text irrelevant to the necessary portion of logic arguments, referring to them as Filler Tex. Filler text lengths are selected from predefined lengths: 0, 4K, 8K, 16K, 32K, 64K, and 128K tokens.

FACTOR consists of three subsets: Easy, Medium, and Hard. Easy subset consists of symbolic reasoning tasks, where the model evaluated is asked to deduce from variables in "Vx" for integer x. For Medium and Hard, two of context in natural language and contains rich hidden operations from semantics. In all three suites, the model is presented with long chains of variables, and they can only provide correct output when they correctly capture the relationship of all variables that appeared. The rest of the context length is filler text. Computation operators are limited to grade school level similar to GSM8K (Cobbe et al., 2021).

The general trend of FACTOR is shown in Figure 1 (a). We comprehensively evaluate state-of-the-art pre-trained LLMs on FACTOR, where Figure 5 presents model names enumeration. Besides, the aforementioned rag technique only achieves 4.5% accuracy on the realistic tasks of FACTOR for op=5 and 4K, far worse than the full attention LLM counterpart (33%). A snippet of our evaluation is shown in Figure 1 (b). Besides, we also perform explainable mathematical modeling of the performance of various models on FACTOR. To our surprise, we observe that the logarithm of accuracy generally correlates with the task complexity *linearly* across all LLMs evaluated, as shown in Figure 1 (c). Through modeling, we obtain more insightful comparisons between different models.

Moreover, we found that two parameters (slope and y-intercept) used in the linear regression possess explainable meanings and be used as quantitative metrics for describing models' abilities and behaviors. For a given context length, the slope, referred to as **Contextual Decay Factor** (CDF), indicates the rate of degradation of the model when solving increasingly longer context. The y-intercept, or **Contextual Decay Offset** (CDO) captures the model's baseline performance at the given context length. We can separately conclude both the model's reasoning ability and the long-context tracking ability from CDF and CDO. Specifically, benefiting from FACTOR design to isolate complexity and context length, we found that most LLMs generally exhibit the following two patterns.

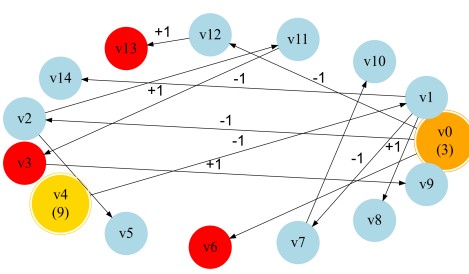 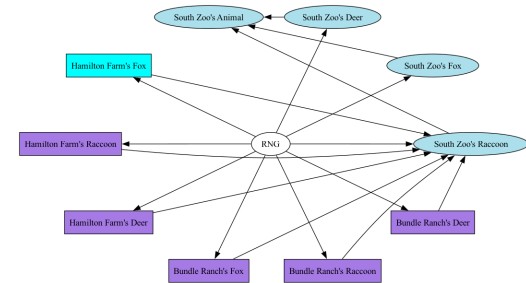

(a) Symbolic Tasks: Variable names in "Vx" for an integer x. Question: which variable is of value y? (for an integer y) Answer in **red**.

(b) Realistic Tasks: Variable names in realistic items that the pretrained LLM is familiar with. Question: How many **Fox** are there in **Hamilton Farm**? Query variable in color **Cyan**.

Figure 2: Examples of FACTOR task. (a) shows the dependence graph of an example for the symbolic portion. In the example, v4 and v6 are initial conditions given, and arrows represent dependencies. For the symbolic tasks, we ask the model to name all variables with a certain value, effectively asking the model to construct the entire dependency graph. (b) shows the dependency graph of an example for the realistic portion. In the example, light cyan surrounding the bounding box is the actual variable we need, while the rest of the blue variables are intermediate variables needed to find before reaching to the query variable, while the purple variables are redundant variables unnecessary for the minimal calculation towards the answer.

- **Decreasing CDO with Increasing Context Length**: Indicates that the model struggles with processing long contexts, declining in its baseline performance regardless of task complexity.

- **Decreasing CDF at Longer Contexts**: Suggests that the model's ability to handle increasing complexity is impaired with longer contexts.

Our contribution can be summarized as follows:

- **Revealing a Log-Linear Accuracy Pattern in Long-Context Reasoning**: In Section 5, we show that the relationship between task accuracy and complexity can be modeled using a simple and consistent log-linear model.

- **Identifying Mechanisms Behind Performance Decay**: Also in Section 5, we uncover two primary mechanisms causing the decay of performance in long-context reasoning: the degradation of logical reasoning ability, quantified by the **Complexity Decay Factor (CDF)**, and the decline in baseline performance with longer contexts, represented by the **Contextual Decay Offset (CDO)**.

- **Reproducing Failure Modes Through Fine-Tuning Strategies**: In Section **??**, our experiments show that the observed failure modes can be reproduced using different fine-tuning methods—*course learning* (as in Llama models) and *mixed sequence length training* (as in GPT-4o-mini). This highlights the impact of training methodologies on models' abilities to handle complex reasoning over long contexts.

- **Unveiling Limitations via Repeated Sampling**: Further in Section **??**, we investigate inference-time strategies like repeated sampling and find that, although they have potential to improve overall performance, inherent biases limit models' abilities to indefinitely extend their reasoning capabilities. The longer the context, the more challenging it becomes to recover performance levels seen with clean context training.

## 2 RELATED WORK

### 2.1 LONG-CONTEXT LANGUAGE MODELS

Various works related to the Long-context Language Model have been proposed. Flash attention(Dao et al., 2022), Flash attention2(Dao, 2024), Ring attention(Liu et al., 2023a), and Tree attention(Shyam et al., 2024) significantly reduced the memory footprint and communication overhead for processing long context in engineering level across multiple nodes. Architectural level innovations such as sparse attentions represented by sliding window attention(Beltagy et al., 2020), are also widely used to reduce the overhead caused by the increasing sequence length. New training strategies, such as gradually extending the training context length in the final stages of pretraining have been applied to support a long context window(Dubey et al., 2024).

## 2.2 LONG CONTEXT BENCHMARKS AND TASKS

There have been a quite a few works benchmarking long-context language models. Existing comprehensive benchmarks like $\infty$bench(Zhang et al., 2024) cover realistic tasks including document QA, summary, and synthetic tasks including information retrieval, expression calculation, extending the context length in the benchmark to over 200k tokens. $\infty$bench(Zhang et al., 2024) does have mathematical reasoning tasks, however the most relevant math.calc part seems to be too difficult for SOTA models to work out. Synthetic tasks often offer more control and are less affected by parametric knowledge in comparison with realistic tasks. One comprehensive synthetic benchmark is RULER(Hsieh et al., 2024), a synthetic benchmark with tasks including retrieval, variable tracking and so on, offering some controls over context length and task complexity. Experiments with various complexities were done, but it does not provide a quantitative analysis of complexity and context length on the correctness of the task, let alone isolate two separate patterns of performance decay. Other benchmarks usually focus on simple retrieval(Github, 2023; Liu et al., 2023b), fact reasoning(Kuratov et al., 2024), the impact of long context on natural language reasoning(Levy et al., 2024) and other real-world knowledge involved tasks.

## 2.3 SYNTHESIZED DATASETS FOR LONG-CONTEXT REASONING

Synthesized tasks are simple to build and absolutely deterministic, data contamination safe, but highly effective to evaluate certain aspects of LLM performance. Its use in long-context benchmarks is profound. Needle-in-the-haystack Kamradt (2023), a pioneering long-context synthesized task, now becomes the go-to task for evaluating LLM long-context retrieval ability. On the other hand, LLM reasoning benchmarks also see recent efforts in synthesized tasks. Mirzadeh et al. (2024) recently proposes to use build synthesized dataset upon GSM8K Cobbe et al. (2021) to study the robustness of LLM reasoning. **Part of our work draws a strong inspiration from a series of works (Ye et al. (2024a), Ye et al. (2024b)) which systematically studies the intricacies of decoder transformers in solving grade-school level problems.** Following their footsteps, we carefully redesign the process of generating the problems so current LLMs can solve without training, and together with thoughtful steps in noise addition, we effectively construct effective reasoning benchmarks for the long-context community.

## 3 RETRIEVAL-AUGMENTED GENERATION VS. LONG CONTEXT BENCHMARKS

In this section, we first show RAG's strong performance on current standard long context LLM benchmarks, and we then show RAG's limitation to grasp the logic sequences and flow on our newly proposed tasks that specifically aim to test the LLM's complex reasoning ability.

We build a standard RAG system for reference. We use MPnet-base-V2 Song et al. (2020) as the sentence retriever and cosine similarity as the metrics for retrieval. For the RAG system decoder, we use Llama-3.1-8B-Instruct. Then, we run the RAG system on standard long context benchmarks in Table 1 and Table 2 with a budget of 2K next to Llama-3.1-8B-Instruct full context. Apart from very minimal tasks, RAG performance on conventional long-context benchmarks is solid and is a close rival to the long-context LLMs.

| Dataset | LLM (0-8k) | LLM + RAG 2K (0-8k) | LLM (>8k) | LLM + RAG 2K (>8k) |
|---|---|---|---|---|
| qasper | 45.86 | 41.40 | 43.65 | 37.97 |
| multifieldqa_en | 52.95 | 51.82 | 48.43 | 43.41 |
| hotpotqa | 56.19 | 56.05 | 44.24 | 47.49 |
| 2wikimqa | 48.05 | 50.41 | 33.40 | 36.01 |
| multi_news | 26.72 | 26.01 | 24.95 | 21.02 |
| triviaqa | 90.85 | 78.60 | 94.36 | 79.60 |
| samsum | 41.88 | 37.68 | 44.49 | 41.95 |

Table 1: Performance comparison of LLM and LLM + RAG 2K across different datasets.

Then, we run the RAG on our tasks. We select the FACTOR medium subset. The RAG cannot solve the task completely. The results are presented in Table 3. We still use 8K context with budget of 2k.

To make the case even more convincing, we now enable iterative prefilling of the RAG system. In other words, we allow the RAG to reselect the context for every token generated. The augmented model then produce 100% accuracy on VT of RULER, which is even higher than 8B decoder in full context. Even though it is better than normal RAG, it still achieves almost 0 accuracy for op > 3, as shown in Table 3. RAG's poor performance

| Models | n_s_1 | n_s_2 | n_s_3 | n_mk_1 | n_mk_2 | n_mk_3 | n_mv |
|---|---|---|---|---|---|---|---|
| Baseline Score | 100.0 | 100.0 | 100.0 | 100.0 | 100.0 | 100.0 | 98.5 |
| RAG Score | 100.0 | 100.0 | 100.0 | 100.0 | 100.0 | 98.0 | 100.0 |

| Models | n_mq | vt | cwe | fwe | qa_1 | qa_2 |
|---|---|---|---|---|---|---|
| Baseline Score | 100.0 | 99.2 | 94.8 | 86.67 | 84.0 | 62.0 |
| RAG Score | 100.0 | 86.4 | 34.8 | 80.67 | 74.0 | 60.0 |

Table 2: RAG on RULER. "n" stands for needle-in-the-haystack, "s" for single, "mk" for multikey, "mv" for multivalue, "mq" for multi-query, "vt" for Variable Tracking.

| op | RAG (with 8B) | Llama 3.1 8B | RAG (with 70B) | Llama 3.1 70B |
|---|---|---|---|---|
| 2 | 0.13 | 0.8024 | 0.175 | 0.975 |
| 3 | 0.11 | 0.5322 | 0.07 | 0.835 |
| 4 | 0.055 | 0.6794 | 0.145 | 0.92 |
| 5 | 0.095 | 0.2641 | 0.1 | 0.87 |
| 6 | 0.08 | 0.2338 | 0.12 | 0.705 |
| 7 | 0.065 | 0.2802 | 0.085 | 0.655 |
| 8 | 0.055 | 0.2459 | 0.05 | 0.59 |
| 9 | 0.05 | 0.1693 | 0.07 | 0.38 |
| 10 | 0.045 | 0.1370 | 0.045 | 0.39 |

Table 3: Performance comparison for RAG and Llama models at different operations.

on FACTOR is due to the high-quality noise we designed to increment context length from essential hundreds of tokens for solving the problem to 8k. We will give more details of the design in Section 4.4.

| Dataset | Iterative Prefill RAG | Llama-3.1-8B-Instruct |
|---|---|---|
| VT from RULER | 100% | 99.2% |
| 2 | 0.52 | 0.8024 |
| 4 | 0.22 | 0.6794 |
| 5 | 0.07 | 0.2641 |
| 6 | 0.02 | 0.2338 |
| 8 | 0.06 | 0.2459 |

Table 4: Comparison of VT from RULER and Noise Type performance between 8K Iterative Prefill RAG and 8K (Llama-3.1-8B-Instruct).

# 4 THE FACTOR BENCHMARK

We introduce the **FACTOR** (**F**actoring **A**nalysis of **C**omplexity and **T**extual **CO**ntext in **R**easoning) benchmark. In the design, we intentionally disentangle the effects of task complexity and context length on language model performance, providing a systematic framework for evaluating reasoning over long contexts. **FACTOR** is subdivided into three different subsets Easy, Medium, and Hard. The task division is classified based on the hierarchical depth of the semantics. The Easy subset consists of assignments and operations acting on symbols, like "Vi" for integer i. Therefore, every operation in this subset is explicit, meaning the model directly follows the instruction to perform operations. More details in 4.1. The medium and hard are questions that have the maximum hierarchical depth of 2 and 3 respectively. Detailed definition and construction in 4.2.

## 4.1 SYMBOLIC PORTION

Tasks are generated by first creating a set of variables $\{v_0, v_1, ..., v_N\}$, where $N$ represents the *task complexity*. Mathematical relationships among these variables (e.g., $v_i = v_j \pm 1$) are then established to form a dependency graph. Consistent values that satisfy all relationships are assigned. These relationships are embedded within filler text, the randomly generated text irrelevant to the logic components, to create contexts of varying lengths, representing different *context lengths*. To distinguish the variable relationships from the filler text, they are

enclosed within triple angle brackets $<<<$ and $>>>$, and further enclosed within @ symbols to separate them from the surrounding content. This ensures that the relationships are clearly identifiable and not affected by the filler text.

To avoid falling back to Variable Tracking-like tasks that use backtracking for answer generation, we choose not to allow the model to calculate only the value of certain query variables. Instead, as shown in the example, we ask the model to output all the variables of a query value, which none can be a valid answer. In practice, our approach works well in differentiating between strong and weak reasoning abilities across models. No partial answer is allowed, so to consistently do the question correctly, the model is asked to traverse through the entire computation graph.

---

**A Symbolic task of FACTOR benchmark**

This is the beginning of the text: $@\langle\langle\langle$ assign $v_1 = v_4 - 1 \rangle\rangle\rangle @@\langle\langle\langle$ assign $v_0 = v_4 - 1 \rangle\rangle\rangle @@\langle\langle\langle$ assign $v_3 = v_4 + 1 \rangle\rangle\rangle @@\langle\langle\langle$ assign $v_2 = 1 \rangle\rangle\rangle @@\langle\langle\langle$ assign $v_4 = v_2 \rangle\rangle\rangle @$ This the end of the text. The text contains relationships between variables enclosed by '$\langle\langle\langle$' and '$\rangle\rangle\rangle$'. These relationships are not sequential assignments in a programming language. They are independent mathematical equations that are all true simultaneously. Using only these relationships, determine what variable(s), if any, are equal to 2. Show your step-by-step reasoning and calculations, and then conclude your final answer in a sentence.
**Answer:** v3

---

## 4.2 COMMONSENSE PORTION

Then, we attempt to map the computation graph into the real-world context. Similar to the symbolic datasets, we strictly control the difficulty level of each problem by the number of binary operations (two variables each) needed to get to the final answer. Our method draws strong inspiration from a previous study Ye et al. (2024a).

**Defining Hierarchical Depth** - The key to designing the commonsense reasoning benchmark similar to Cobbe et al. (2021) is to craft the hidden operations. The difficulty of hidden operations is controlled by the hierarchical structure of the context. For example, the context of "Animals in the Location" contains two classes of objects ("Animals" and "Location") where Animals is the possession of Location. It is a structure with hierarchical depth two. Given a context of two layers of objects, we have the most fundamental hidden operation: addition. For example, "The number of Beverly Forest's Fox is 2. The number of Beverly Forest's Wolves is 3. Assuming there isn't any other type of animal in Beverly Forest." The total number of animals in Beverly Forest equals the sum of the first two objects, even though there isn't any description of addition in the text. Context of higher hierarchical depth usually contains all possible relationships of the previous plus the newly emerged relationship pattern, making them harder to deduce.

On the other hand, adding "number of children" to the previous context increments the hierarchical depth to three. Continuing the previous sentence, "The average number of children of Beverly Forest's Wolves is 3, while the average number of children of Beverly Forest's Fox is 2. What is the total number of Beverly Forest's animal children?" To compute that we need to multiply each animal count with their average count of children and sum up all the animals. Similarly, the chain of operations isn't narrated explicitly in the text, but it is intuitive for humans to apply. As an example, for depth of three, it contains the direct operations in the Easy subset as well as the addition hidden operation, plus the sum of multiply. We naturally designate problems with a depth of two to be in the Medium subset, and a depth of three to be in the Hard subset. Below we present an example with FACTOR HARD where blue signifies abstract variables that appear starting from Medium (depth 2), whereas cyan signifies abstract variables that only appear starting from Hard (depth 3). Commonsense reasoning problems mostly contain depth 2 and depth 3 relationships (Ye et al. (2024a)), where deeper hierarchy is rare even in normal natural language.

**Making the solution** - Similar to the Easy subset, we start from the randomly generated computational graph, which is a DAG, and then attach numbers, forming abstract parameters to the graph. Out from the computational graph, we randomly select a variable as the query variable, then perform a topological sort of the computation graph. For the nodes on the topological sort list that aren't pointed to by other variables, we initialize their initial values and continue going through the sort list until the query variable. The entire chain is the solution, while the value of the query variable is the answer. Below we present a full example of FACTOR Hard.

> **A typical problem of FACTOR Hard**
>
> Problem: The number of adult deer in Oakridge Riverside equals 2 times the total number of newborn animal children in Cedar Valley. The number of adult deer in Cedar Valley equals 2. The average number of newborn children per adult deer in Oakridge Riverside equals 4. The average number of newborn children per adult deer in Cedar Valley equals the number of adult deer in Cedar Valley. Question: What is the total number of adult animals in Oakridge Riverside?
> **Solution**: Define adult deer in Cedar Valley as s; so s = 2. Define average number of newborn children per adult deer in Cedar Valley as i; so i = s = 2. Define total number of newborn animal children in Cedar Valley as M; so M = s * i = 2 * 2 = 4. Define adult deer in Oakridge Riverside as h; z = M = 4; so h = 2 * z = 2 * 4 = 8. Define total number of adult animals in Oakridge Riverside as Z; so Z = h = 8. Answer: 8.

### 4.3 REVERSE REASONING PATH

Our problem construction mandates the use of a topological sort list, so all the hidden operations we added are constructive, always either addition or multiplication. However, to enable hidden operations of subtraction and division naturally through our generation pipeline, we also designate half of the problem in the benchmark to be of the reverse reasoning path. For the reverse reasoning path, we first still construct the topological list first. Then we initialize the previous query variable at the end of the sort list. Then, we assign the other side of the sorted list, variables that no other nodes point to as the query variable. Therefore, since we go in the complete opposite direction of the normal forwarding logic consisting of addition and multiplication, we now have natural subtraction and division.

**Overview of Commonsense subsets** - For both the medium and hard subsets, we have half of the problem in the normal forward, and half in the reverse path. Also, we deploy three different contextual templates to diversify the problem text. These templates are "location-animal-children", "city-school-teacher", and "festival-movie-nomination".

### 4.4 NOISE ADDITION

Adding noise is essential to enriching the length of the problem in the long context regime. For FACTOR, we mainly explore two different directions of making the noise. For FACTOR Easy, we follow conventional long context benchmarks and generate random noise to increase the context length. On the other hand, for FACTOR Medium and Hard, Benefiting from our commonsense reasoning problem generator, we can generate noise statements that are in the same format and semantics as the essential logic statements. In practice, we also encourage the noise statements to be noise variables pointed by essential variables to enlarge the connection between the core graph and the noise graph. This tight connection is the key to why RAG cannot solve the problem effectively. On the other hand, long context LLM isn't sensitive to close noise.

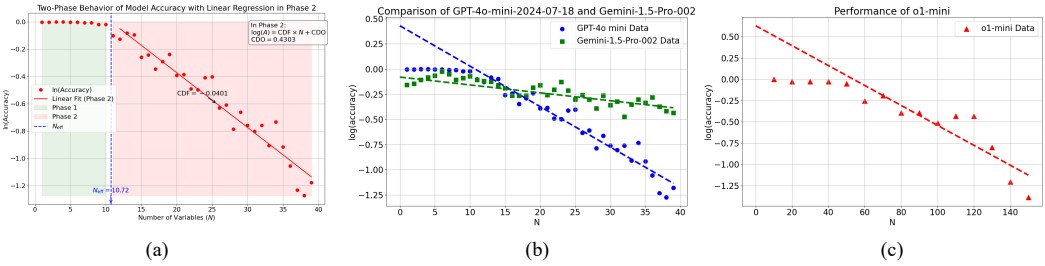

(a)                                        (b)                                        (c)

Figure 3: (a) Illustration of the two-phase accuracy behavior as a function of task complexity $N$. (b) presents the two different accuracy patterns between GPT-4o-mini and Gemini-1.5-Pro. GPT-4o-mini outperforms Gemini-1.5-Pro in low complexity tasks, while Gemini-1.5-Pro is more capable of dealing with more complex tasks. (c) shows the amazing performance of o1-mini: its has a higher $N_{eff}$ than any other models tested and a decent CDF. This illustrates the effectiveness of the inference time strategy on reasoning.

## 5 EVALUATION ON PRETRAINED MODELS

We evaluate a range of pre-trained language models using the FACTOR benchmark to understand how they handle increasing task complexity and long context lengths, identifying their failure modes. The evaluation

is structured into two parts: (1) **Benchmarking with Zero Filler Context**: Assessing models' abilities to handle task complexity independently of context length. (2) **Benchmarking with Long Contexts**: Analyzing how models that perform well without filler context degrade when exposed to long contexts. **Here due to page limit, we mainly focus on the FACTOR Easy task, we put the evaluation on the Medium and Hard tasks later in the Appendix.**

## 5.1 EVALUATION METRICS

Besides just getting the accuracy, we also care about modeling the relationships between the accuracy and the complexity, thanks to the FACTOR design to isolate complexity and context length.

**Two-Phase Accuracy Behavior** - Models generally exhibit a characteristic two-phase behavior (Figure 3 (a)) in accuracy as the number of variables $N$ increases (see Figure 3). **Phase 1**: For small values of $N$, models maintain near-perfect accuracy, effectively handling tasks with low complexity. **Phase 2**: Beyond a critical complexity threshold ($N_{\text{eff}}$), accuracy declines exponentially with increasing $N$, indicating rapid degradation in performance for more complex tasks. This pattern suggests a limit to the task complexity that models can handle before performance significantly deteriorates to close zero, where it plateaus.

It is intuitive to fit the accuracy versus operations using logistic regression that can fit all two phases elegantly. Also, another candidate is a stepwise function with a constant for phase 1 following a exponential decay function. The only difference is at the transition between the two phases, where the logistic regression predicts a smooth transition, while exponential decay is a sharper one. Later, these two functions are mathematically similar, even close to identical when x gets better. We compare these two methods rigorously on O1-mini's behavior, where we found that the step-wise gives 0.006 MSE score from ops 0-150, compared with 0.013 from logistic regression, suggesting that the true transition from top LLM is sharp. On the other hand, due to lack of reasoning ability, most LLMs barely have the phase 1 displayed. Therefore, the focus of our analysis now is on the decay, where the two candidates are very similar. We proceed with exponential decay function as the modeling tool for later analysis because of its surprisingly strong vicinity to our collected data. Extensive studies and fitting to justify using exponential function is presented in Appendix B.

**Evaluation Metrics Definition** - In Phase 2, accuracy $A$ decreases exponentially with increasing $N$. By taking the natural logarithm of accuracy, we linearize this decay:

$$\log(A) = \text{CDF} \times N + \text{CDO}$$

Where **Complexity Decay Factor (CDF)** is the negative slope of the line (CDF $< 0$), representing the rate at which accuracy decays with increasing task complexity; **Contextual Decay Offset (CDO)** is the intercept of the line, capturing the baseline performance level influenced by context length. Our two-phase model explains well the pattern of decreasing model accuracy with task complexity, and this model does not predict accuracy outside the 0-1 range.

From these parameters, we define the **Effective Complexity** $N_{\text{eff}}$, indicating the maximum task complexity the model can handle before significant performance degradation. It is calculated as the value of $N$ when the extrapolated $\log(A) = 0$ (i.e., when accuracy $A = 1$): $N_{\text{eff}} = -\frac{\text{CDO}}{\text{CDF}}$. However, if the CDO is negative, the extrapolated $N_{\text{eff}}$ becomes negative, which is not meaningful since we cannot have a negative number of variables. In such cases, the two-phase behavior is not observed—the model's accuracy declines from the outset without an initial phase of high accuracy. Therefore, for negative CDO values, we focus on the exponential decay characterized by the CDF and CDO. A more detailed description of data processing can be found in the Appendix D.

## 5.2 SYMBOLIC TASKS: BENCHMARKING WITH ZERO FILLER CONTEXT

We evaluated 15 models, comparing their FACTOR benchmark metrics with ELO scores from the LMSYS Chatbot Arena(Chiang et al., 2024) on hard prompts with style control (see Table 5). We use the area under the accuracy curve (AUC) to rank the models. In particular, AUC40 denotes the AUC calculated for N less than 40.

The evaluation reveals key insights into model performance concerning task complexity: Models with similar LMSYS ELO scores exhibit different behaviors as task complexity $N$ increases(see Figure **??**): models like **GPT-4o-mini-2024-07-18** (CDO = 0.4303, CDF = $-0.0401$) excel on simple tasks (high CDO) but degrade rapidly with increasing complexity (more negative CDF); Models like **Gemini-1.5-Pro-002**(Team et al., 2024) (CDO = $-0.0696$, CDF = $-0.0081$) struggle with simple tasks (low CDO) but handle complex tasks better as they degrade more slowly (less negative CDF). **o1-mini** achieves both high CDO (0.6303) and a less negative CDF ($-0.0117$), resulting in a high $N_{\text{eff}}$ (53.87). This indicates that inference-time strategies can significantly enhance performance across task complexities.(see Figure 3)

Table 5: Metrics with Zero Filler Context are listed in this table. We observe a strong correlation between model capability and these metrics, together with different accuracy patterns of different models.

| Index | Model | CDF | CDO | $N_{\text{eff}}$ | AUC | AUC40 | ELO |
|---|---|---|---|---|---|---|---|
| 1 | o1-mini | -0.0117 | 0.6303 | 53.87 | 139.34 | 38.25 | 1294 |
| 2 | Gemini-1.5-Pro-002 | -0.0081 | -0.0696 | -8.59 | 114.87 | 31.42 | — |
| 3 | GPT-4o-2024-05-13 | -0.0220 | 0.2298 | 10.45 | 55.90 | 31.31 | 1251 |
| 4 | GPT-4o-2024-08-06 | -0.0205 | 0.0899 | 4.39 | 53.17 | 29.45 | 1237 |
| 5 | Claude-3-5-Sonnet-20240620 | -0.0187 | -0.1447 | -7.74 | 46.27 | 25.52 | 1268 |
| 6 | Qwen2.5-72B-Instruct | -0.0265 | 0.2056 | 7.76 | 45.50 | 28.26 | 1223 |
| 7 | Mistral-Large-Instruct-2407 | -0.0279 | 0.2100 | 7.53 | 43.37 | 28.12 | 1231 |
| 8 | Gemini-1.5-Flash-002 | -0.0244 | -0.0180 | -0.74 | 40.25 | 25.04 | — |
| 9 | GPT-4o-mini-2024-07-18 | -0.0401 | 0.4303 | 10.73 | 35.67 | 27.19 | 1219 |
| 10 | GPT-4-Turbo-2024-04-09 | -0.0378 | 0.2514 | 6.65 | 33.10 | 25.06 | 1226 |
| 11 | Llama-3.1-70B-Instruct | -0.0302 | -0.0481 | -1.59 | 31.56 | 21.6 | 1187 |
| 12 | Qwen2-72B-Instruct | -0.0467 | 0.0123 | 0.26 | 21.67 | 17.91 | 1178 |
| 13 | Claude-3-Haiku-20240307 | -0.0471 | -0.0848 | -1.80 | 19.50 | 16.87 | 1173 |
| 14 | Mistral-Nemo-Instruct-2407 | -0.0608 | 0.0735 | 1.21 | 17.66 | 15.85 | — |
| 15 | Llama-3.1-8B-Instruct | -0.0694 | -0.4615 | -6.65 | 9.08 | 8.10 | 1132 |

Models may perform similarly on general benchmarks but differ on tasks requiring complex reasoning. Selecting models for applications should consider CDO and CDF to match the complexity needs. The FACTOR benchmark highlights the necessity to evaluate models on complexity handling rather than solely on overall scores. By focusing on CDO and CDF, we are able to model distinct failure modes among models as task complexity increases. Understanding these metrics aids in selecting appropriate models for specific tasks and emphasizes the importance of specialized benchmarks like FACTOR.

## 5.3 SYMBOLIC TASKS: BENCHMARKING WITH LONG CONTEXTS

We analyze how models degrade when exposed to long contexts by examining the CDF and CDO metrics across different context lengths. Models were tested with varying amounts of filler context, extending up to their maximum context lengths (4K to 128K tokens). (see Table 6 and 7)

Table 6: Complexity Decay Factor (CDF) for Models at Different Context Lengths. (Values scaled by $10^2$)

| Model | 4K | 8K | 16K | 32K | 64K | 128K |
|---|---|---|---|---|---|---|
| **Mistral-Large-Instruct-2407** | −3.26 | −3.88 | −4.04 | −4.41 | −15.19 | — |
| **Qwen2.5-72B-Instruct** | −2.82 | −3.16 | −3.27 | −3.08 | — | — |
| **GPT-4o-mini-2024-07-18** | −4.93 | −4.90 | −4.95 | −5.13 | −5.01 | −6.02 |
| **Llama-3.1-70B-Instruct** | −3.88 | −4.27 | −4.82 | −5.55 | — | — |
| **Qwen2-72B-Instruct** | −4.66 | −5.22 | −5.56 | −6.10 | — | — |
| **Mistral-Nemo-Instruct-2407** | −5.15 | −6.58 | −29.43 | −4.35 | −3.58 | −3.64 |
| **Llama-3.1-8B-Instruct** | −7.24 | −7.12 | −8.48 | −9.98 | −10.19 | — |

The evaluation reveals distinct failure modes when models are exposed to longer contexts.(all CDF values are scaled by $10^2$, see Figure 4) **(1) Stable CDO, Degrading CDF (e.g., Llama Series):** Models like **Llama-3.1-70B-Instruct** maintain relatively stable CDO across context lengths (from −0.014 at 4K to −0.104 at 32K), indicating consistent baseline performance. However, their CDF becomes more negative as context length increases (from −3.88 to −5.55), signifying increased difficulty with task complexity in longer contexts. **2. Stable CDF, Degrading CDO (e.g., GPT-4o-mini-2024-07-18):** models like **GPT-4o-mini-2024-07-18** maintains a consistent CDF across context lengths (approximately −4.93 to −5.13 up to 64K tokens), suggesting stable handling of task complexity. However, its CDO decreases steadily (from −0.020 at 4K to −0.711 at 64K), indicating declining baseline performance with longer contexts. **3. Degrading CDO and CDF:** models like **Mistral-Large-Instruct-2407** show degradation in both CDO and CDF as context length increases, facing compounded difficulties with baseline performance and task complexity.

Table 7: Contextual Decay Offset (CDO) for Models at Different Context Lengths.

| Model | 4K | 8K | 16K | 32K | 64K | 128K |
|---|---|---|---|---|---|---|
| **Mistral-Large-Instruct-2407** | 0.085 | −0.027 | −0.153 | −0.497 | −0.510 | — |
| **Qwen2.5-72B-Instruct** | 0.055 | −0.021 | −0.088 | −0.281 | — | — |
| **GPT-4o-mini-2024-07-18** | −0.020 | −0.212 | −0.312 | −0.494 | −0.711 | −0.700 |
| **Llama-3.1-70B-Instruct** | −0.014 | −0.065 | −0.054 | −0.104 | — | — |
| **Qwen2-72B-Instruct** | −0.172 | −0.210 | −0.197 | −0.296 | — | — |
| **Mistral-Nemo-Instruct-2407** | −0.459 | −0.609 | −0.482 | −1.154 | −1.287 | −1.287 |
| **Llama-3.1-8B-Instruct** | −0.591 | −0.679 | −0.686 | −0.608 | −0.621 | — |

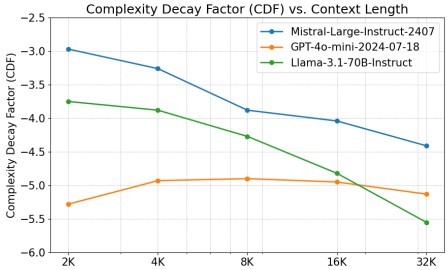 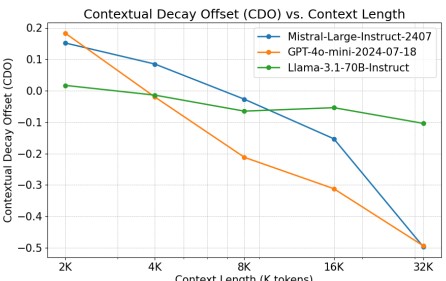

Figure 4: CDF/CDO of three models at different context length. Llama-3.1-70B-Instruct maintains a relatively stable CDO across context lengths, while its CDF decreases over context length. GPT-4o-mini, on the contrary, maintains a relatively stable CDF while its CDO decreases over context length. Mistral-Large-Instruct is observed to have both metrics decreasing over context length.

Models exhibit different failure modes with long contexts: (1) **Stable CDO, Degrading CDF**: Models maintain baseline performance but increasingly struggle with task complexity as context lengthens; (2) **Stable CDF, Degrading CDO**: Models handle task complexity consistently but suffer declining overall performance with longer contexts; (3) **Degrading CDO and CDF**: Models face difficulties with both baseline performance and task complexity. Understanding these patterns is crucial for selecting appropriate models based on task requirements and highlights the importance of enhancing model robustness in processing long contexts and complex tasks.

## 6 CONCLUSION

In this paper, we introduced the FACTOR benchmark, a novel framework designed to systematically evaluate the complex reasoning abilities of large language models (LLMs) over long contexts. A key innovation of our work is the modeling of performance over task complexity, moving beyond traditional scalar evaluation metrics to capture the dynamic two-phase behavior in accuracy as complexity increases. Specifically, we observed that LLMs maintain high accuracy up to a certain complexity threshold, after which performance declines exponentially. By characterizing this behavior through the **Complexity Decay Factor (CDF)** and the **Contextual Decay Offset (CDO)**, we provided a nuanced understanding of how task complexity and context length independently affect model performance.

Our analysis revealed that these metrics not only quantify the degradation of logical reasoning ability and baseline accuracy but also highlight the limitations of current LLMs in handling complex reasoning tasks over extended contexts. Furthermore, we demonstrated that different fine-tuning strategies can reproduce these failure modes, emphasizing the significant impact of training methodologies on model capabilities. By modeling the two-phase behavior in accuracy rather than relying on a single performance score, the FACTOR benchmark offers a more detailed and insightful evaluation framework. This approach allows researchers to identify specific areas for improvement in LLMs and guides future developments in creating more robust language models capable of complex reasoning over long contexts.

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

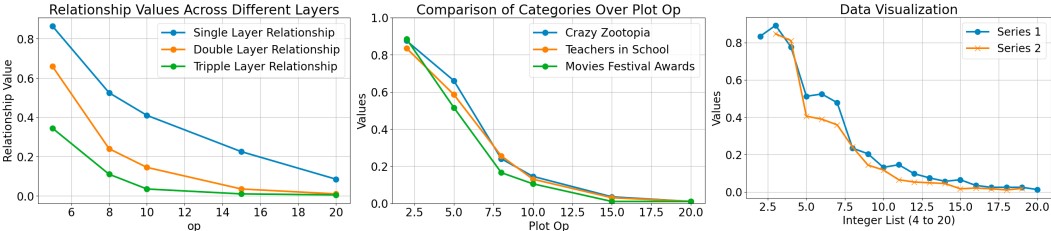

Figure 5: Left shows the difference between hierarchical depth in LLM evaluation under zero-context; Middle shows the comparison between different templates, showing they are the same; Right shows the performance comparison between forward logic and reverse logic

.

# A    MEDIUM AND HARD SUBSETS

In this section, we present more results on Medium and Hard subsets of FACTOR. We first compare all three different subsets in A.1. We then present more results on Llama-3.1-8B-Instruct and Llama-3.1-70B-Instruct on these two subsets in A.2. We compare forward reasoning and reverse reasoning in A.3. Comparison with different templates in A.4.

## A.1    EASY VS. MEDIUM VS. HARD

First, we compare the Easy, Medium, and Hard subsets in the FACTOR dataset. The results are summarized in Table 8. You can see that from easy to medium to hard, the accuracy curve decays progressively quickly. The difference between Easy, Medium, and Hard is essentially the difference in hierarchical depth. To study the effect of hierarchical depth, we also make the depth-1 version of the commonsense problem generator. We plot them together in Figure 5 (left).

| op | Easy | Medium | Hard |
|----|------|--------|------|
| 5 | 0.808 | 0.86 | 0.7 |
| 8 | 0.832 | 0.875 | 0.512 |
| 10 | 0.708 | 0.612 | 0.41 |
| 12 | 0.628 | 0.56 | 0.36 |
| 15 | 0.556 | 0.43 | 0.19 |
| 20 | 0.516 | 0.2 | 0.05 |

Table 8: Performance comparison across difficulty levels (Easy, Medium, Hard) for different operations.

## A.2    FULL RESULTS

| Models | Zero medium | Zero hard | 4k medium | 4k hard | 8k medium | 8k hard |
|--------|-------------|-----------|-----------|---------|-----------|---------|
| Llama-3.1-8B-Instruct | 466.6 | 302.2 | 310.0 | 223.5 | 201.3 | 152.2 |
| Llama-3.1-70B-Instruct | 1000.7 | 736.5 | 769.7 | 699.5 | 345.9 | 355.2 |

| Models | 16k medium | 16k hard | 32k medium | 32k hard |
|--------|------------|----------|------------|----------|
| Llama-3.1-8B-Instruct | 174.3 | 129.9 | 172.4 | 113.1 |
| Llama-3.1-70B-Instruct | 253.3 | 234.2 | 215.1 | 169.6 |

Table 9: Comparison of models AUC across different context lengths. For zero-context and 4k, the AUC is computed between 2 and 20 inclusively, while the rest uses 2 to 10, because after then, they are very close to zero

Due to the main body's space limitations, we place the results of the commonsense reasoning Medium and Hard subsets of FACTOR here. Currently, Llama-3.1-8B-Instruct and Llama-3.1-70B-Instruct are thoroughly evaluated. We found that the trend we observed on FACTOR Easy also holds for FACTOR Medium and Hard. As a case study, we plot the FACTOR medium in Figure 6 (a) and (b). The decaying trend is also

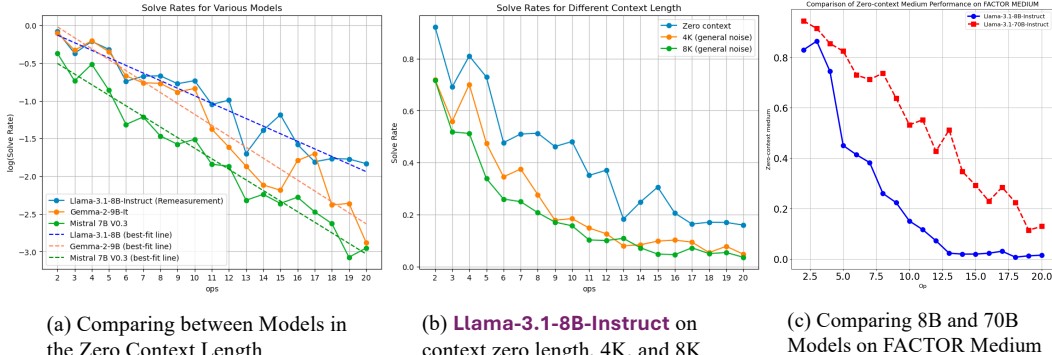

(a) Comparing between Models in the Zero Context Length

(b) **Llama-3.1-8B-Instruct** on context zero length, 4K, and 8K

(c) Comparing 8B and 70B Models on FACTOR Medium

Figure 6: Realistic Portion Results. (a) shows that the log-linear relationship also holds true for more complex datasets as well across three different LLMs. (b) shows that the performance generally degrades as the context length increases. (c) shows that for two different noise addition methods, which shows that the two noise aren't much different to LLMs across both 4K and 8K context length.

exponential, and if we take the accuracy logarithm, we see that the linear curve can also fit it neatly. We also summarize the AUC score for both 8B and 70B. For both medium and hard subsets and each ops split, we have 200 problems of forward logic, and 200 problems of reverse logic, as described in 4.3. We can compare the 8B and 70B models on FACTOR Medium zero context. We can see that the 8B model plateau way faster than the 70B model. Therefore, FACTOR potentially offers a much more comprehensive evaluation of LLMs performance in a easy to scale up difficulty manner.

## A.3  FORWARD VS. REVERSE

Image shown in Figure 5 (right). The forward and backward generations are run using the 8B model. We can see that overall two curves have a similar trend. The reverse (yellow) is consistently less than or on par (not surpass).

## A.4  TEMPLATE

Figure 5 (right) shows the comparison between the three contextual template. The dataset is collected for Llama-3.1-8B-Instruct under zero-context. The difference might be subtle and small, which supports the decision to use multiple templaThe forward reasoning is slightly better for LLM to reverse logic. Adding multiple templates is leading huge improvement in text diversity.

## B  VISUALIZATION OF MODELING EXPONENTIAL DECAYINGJ CURVES

|  | Loglinear Fit | Logistic Regression Fit |
|---|---|---|
| O1-mini on FACTOR easy | 0.01349 | 0.006215 |
| Llama-3.1-8B-Instruct on FACTOR easy | 0.003196 | 0.026159 |
| Llama-3.1-70B-Instruct on FACTOR-easy | 0.002697 | 0.041390 |
| Llama-3.1-8B-Instruct on FACTOR medium | 0.001932 | 0.003372 |
| Llama-3.1-70B-Instruct on FACTOR medium | 0.004230 | 0.016529 |

Table 10: Comparison of Loglinear Fit and Logistic Regression Fit across different models and datasets.

The experiment results show that loglinear is a strong modeling method, better than logistic regression.

We also show a study with much higher precision in Figure 8 on Gemini-1.5-Flash. We only focus on the decaying and then plateauing at near-zero performance. We fit logistic regression fit in red, but exponential decay in blue. We report MSE in the legends, we can see that exponential fit does provide a better fit, because of its sharp decrease.

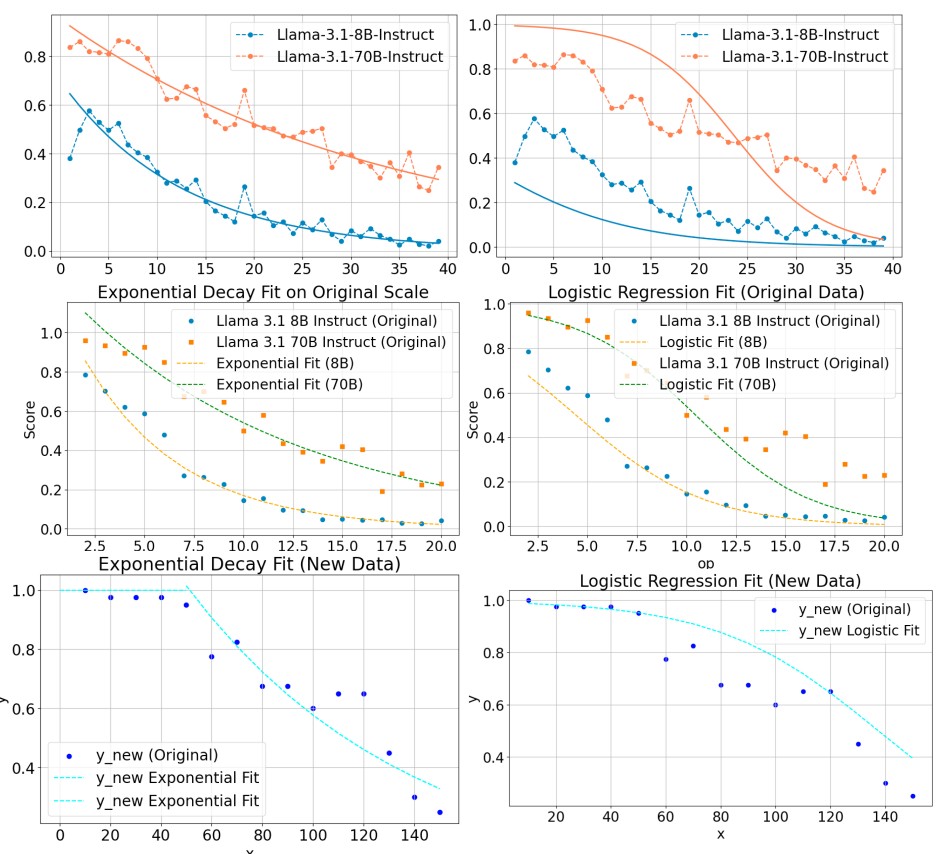

Figure 7: Comparing Between Logistic Regression and LogLinear fit, Loglinear offers much closer fit to the plotted scatter points

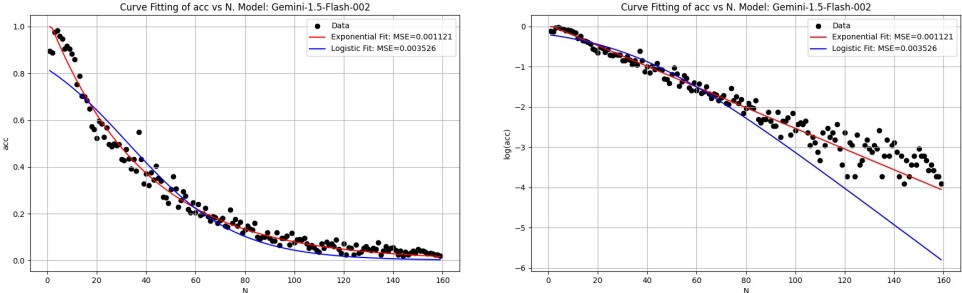

Figure 8: High precision fitting study on the decaying phase and near zero plateauing phases of Gemini-1.5-Flash. Logistic regression fit and Exponential decaying curve are both presented in red and blue respectively. The exponential decaying curve fits the high-precision scatter plots better according to the MSE score

## C  TASK GENERATION AND DISTRIBUTION

In this section, we provide a detailed explanation of the process used to generate the synthetic tasks in the FACTOR benchmark. These tasks are meticulously designed to evaluate the reasoning abilities of language models over varying levels of task complexity and context length while ensuring that these two factors are independently controlled.

## C.1 TASK GENERATION PROCESS

The generation of each task involves several steps: creating variables and relationships, generating the payload, preparing the context, inserting the payload into the context, and forming the question prompt. This process is carefully constructed to introduce variability and prevent models from relying on simple heuristics or memorization.

### C.1.1 VARIABLE AND RELATIONSHIP CREATION

To control task complexity, we vary the number of variables $N$, where each variable is denoted as $v_i$ for $i = 0, 1, ..., N-1$. We establish interdependent relationships among these variables by generating a directed forest—a collection of trees representing dependencies.

We start by randomly shuffling the variables to introduce variability. For each variable $v_i$ where $i \geq k$ (with $k$ being a randomly selected parameter between 1 and $N$), we randomly select a parent variable $v_p$ from the set $\{v_0, v_1, ..., v_{i-1}\}$. This process creates directed edges from $v_p$ to $v_i$, establishing dependency relationships.

To define the mathematical relationships between the variables, we assign simple operations to each edge. These operations are randomly chosen from $\{$no operation$, +1, -1\}$. The use of these simple operations ensures that the calculations required to solve the tasks remain within the realm of basic arithmetic, preventing models from facing overly complex computations.

**Variable Value Assignment**  After establishing the relationships, we assign integer values to the variables while satisfying the dependencies. For root variables (those without parents), we assign random integer values between 0 and 10. This range is chosen to keep the numerical values small, again to prevent the need for complicated calculations.

For each child variable $v_i$, its value is computed based on its parent's value and the assigned operation:

$$\text{value}(v_i) = \begin{cases} \text{value}(v_p) + 1, & \text{if the operation is } +1 \\ \text{value}(v_p) - 1, & \text{if the operation is } -1 \\ \text{value}(v_p), & \text{if there is no operation} \end{cases}$$

This approach maintains simplicity in the calculations required, ensuring that the tasks assess the models' reasoning abilities rather than computational prowess.

### C.1.2 PAYLOAD GENERATION

The payload consists of the variable relationships formatted as textual statements. Each relationship is expressed as an assignment statement, enclosed within triple angle brackets `<<<` and `>>>` to clearly distinguish them from the filler text. These statements are further enclosed within single `@` symbols when inserted into the context.

An example of a payload statement is:

$$\texttt{<<<assign } v_i = v_p \text{ [operation]>>>}$$

where `[operation]` is either `+ 1`, `- 1`, or left empty if there is no operation. The payload statements are shuffled to present the relationships in a non-sequential order, adding to task complexity by preventing models from relying on the order of presentation.

### C.1.3 CONTEXT PREPARATION

To control the context length independently, we generate filler text of specified lengths. The filler text is irrelevant to the variable relationships and serves to increase the context length, simulating scenarios where critical information is embedded within large amounts of unrelated data.

The filler text is created by randomly selecting words from a predefined list related to computational topics, such as "algorithm," "data," "performance," and so on. These words are concatenated to form sentences, with occasional punctuation added to simulate natural language text. The filler text may also include random sentences or phrases inserted between the payload statements to further obscure the relationships and mimic real-world text where key information is interleaved with irrelevant content.

### C.1.4 PAYLOAD INSERTION

The payload of variable relationships is inserted into the filler text at random positions. The filler text is first tokenized into sentences using the NLTK library's sentence tokenizer. Insertion points are determined based on specified intervals, ensuring that the payloads are dispersed throughout the context rather than clustered together.

Each payload statement, enclosed within @ symbols and <<<>>> brackets, is inserted at the selected points. The maximum group size parameter controls how many payload statements can be inserted together at a single insertion point, adding another layer of variability. Additionally, filler content may be placed between payload statements, further increasing the challenge by requiring the model to discriminate between relevant and irrelevant information.

### C.1.5 QUESTION PROMPT FORMATION

Finally, we generate the question prompt that instructs the model on the task to perform. The prompt includes:

- Delimiters indicating the beginning and end of the text. - An explanation that the relationships enclosed by <<< and >>> are independent mathematical equations that are all true simultaneously, and not sequential assignments in a programming language. - A task instruction, asking the model to determine which variable(s), if any, are equal to a randomly selected target value. The target value is chosen from among the variable values or from values slightly outside the range of assigned values, which may result in no variables matching the target, thereby introducing unpredictability. - A request for the model to show step-by-step reasoning and conclude with the final answer in a sentence.

This structure ensures that the model must process and reason over the entire context, filtering out irrelevant information and correctly interpreting the relationships to arrive at the answer.

## C.2 TASK DISTRIBUTION

We generate tasks across a wide range of complexities and context lengths to create a comprehensive evaluation suite.

### C.2.1 COMPLEXITY LEVELS

The number of variables $N$ varies from 1 to 39, covering a broad spectrum of task complexities. For each value of $N$, we generate multiple tasks with different configurations to ensure diversity. The parameter $k$, controlling the number of trees in the forest, is randomly selected for each task and ranges from 1 to $N$.

### C.2.2 CONTEXT LENGTHS

Context lengths are set to predefined values: 0 (no filler text), 1K, 2K, 4K, 8K, 16K, 32K, 64K, and 128K tokens. Our benchmark in the main context only include subsets of 0 and 4K or more. For each context length, filler text is generated accordingly, and the payloads are inserted as described. The inclusion of different context lengths allows us to evaluate how models handle tasks when key information is embedded within varying amounts of irrelevant text.

### C.2.3 SAMPLE SIZES

For each combination of $N$ and context length, we generate 50 distinct task instances. This results in a total of 1,950 tasks for each context length setting (39 values of $N$ times 50 tasks). The large sample size ensures that our evaluation is statistically robust and that any observed trends are not due to random chance.

## C.3 EXAMPLE TASK

An example of a generated task is as follows:

---

**Sample Task**

**Context:**
This is the beginning of the text:
@<<<assign $v_1 = v_4 - 1$>>>@ *algorithm data optimization.* @<<<assign $v_0 = v_4 - 1$>>>@ *performance analysis.* @<<<assign $v_3 = v_4 + 1$>>>@ *code snippet.* @<<<assign $v_2 = 1$>>>@ *best practice.* @<<<assign $v_4 = v_2$>>>@
[...filler text continues...]
This is the end of the text.
The text contains relationships between variables enclosed by '<<<' and '>>>'. These relationships are not sequential assignments in a programming language; they are independent mathematical equations that are all true simultaneously.
Using only these relationships, determine what variable(s), if any, are equal to 2. Show your step-by-step reasoning and calculations, and then conclude your final answer in a sentence.
**Answer:**
v_3

---

In this example, the variables $v_0$ to $v_4$ have interdependent relationships defined by the equations provided. Filler content, such as "algorithm data optimization" and "performance analysis," is interspersed between the payload statements, increasing the context length and complexity. The task requires the model to:

- Extract the relevant variable relationships from the payloads. - Interpret and solve the system of equations. - Determine which variable(s) equal the target value (in this case, 2). - Present the reasoning and final answer in a coherent manner.

This example illustrates how the task assesses the model's ability to perform multi-step reasoning over extended contexts that include irrelevant information.

### C.4 ENSURING DIVERSITY AND AVOIDING DATA LEAKAGE

To prevent models from exploiting patterns or memorizing specific instances, we employ several strategies:

- Variable names are randomly shuffled for each task instance. - Relationships are presented in a random order and interleaved with filler content. - Operations assigned to edges are randomly selected from simple options to maintain calculation simplicity while adding variability. - Target values for the query are randomly chosen and may not correspond to any variable value in the task, introducing the possibility that no variables meet the condition.

These measures create a wide variety of task instances, reducing the likelihood of data leakage or overfitting. Models must genuinely understand and reason through each task rather than relying on memorization or pattern recognition.

### C.5 SUMMARY

The task generation process in the FACTOR benchmark is carefully designed to independently control task complexity and context length while preventing models from relying on shortcuts. By varying the number of variables and their interdependencies, we manipulate task complexity. By inserting filler text of specified lengths and interleaving filler content between payload statements, we manipulate context length and complexity. The use of simple operations and small integer values ensures that the focus remains on reasoning rather than computation.

This systematic approach allows us to create a diverse set of tasks that robustly evaluate models' reasoning abilities over long contexts. By disentangling the effects of task complexity and context length on model performance, the FACTOR benchmark facilitates a deeper understanding of models' strengths and limitations in reasoning over extended textual inputs, guiding future improvements in language model development.

## D DATA ANALYSIS AND LINEAR REGRESSION METHODOLOGY

This section details the methodology used for the linear regression analyses in our study, including data selection criteria, regression techniques, handling of low accuracy values, and confidence interval calculations. The approach aligns with the code used in our interactive analysis.

### D.1 TWO-PHASE ACCURACY BEHAVIOR AND DATA SELECTION

Our experiments reveal a characteristic two-phase accuracy behavior as task complexity $N$ increases:

(1) **Phase 1**: High accuracy plateau where models perform well on simpler tasks, typically with accuracies close to 100%.

(2) **Phase 2**: Exponential decay of accuracy as task complexity exceeds a certain threshold $N_{\text{eff}}$.

To model the rate of accuracy decline in Phase 2, we perform linear regression on the natural logarithm of accuracy versus task complexity $N$. It is essential to focus on data from Phase 2 only, to avoid distortion from the Phase 1 plateau.

### D.2 DATA SELECTION CRITERIA

Selecting appropriate data ranges is crucial for accurate regression. We use different accuracy ranges for different models and experimental stages to focus on Phase 2 data:

(1) **Pretrained Models**: Accuracy between $[0.1, 0.9]$. (2) **Train-from-Scratch Models (Pretraining Stage)**: Accuracy between $[0.1, 0.8]$. (3) **Fine-tuned Models**: Accuracy between $[0.2, 0.8]$. (4)**Repeated Sampling Experiments**: Accuracy between $[0.02, 0.8]$.

These ranges exclude Phase 1 data (high accuracy plateau) and avoid extremely low accuracies that can lead to numerical instability.

### D.3 LINEAR REGRESSION METHODOLOGY

For each model and experimental condition, we perform linear regression to fit:
$$\log(A) = aN + b$$
where:

- $A$ is the accuracy for task complexity $N$.

- $a$ is the **Complexity Decay Factor (CDF)**, indicating the rate of exponential decay.

- $b$ is the **Contextual Decay Offset (CDO)**, representing baseline performance at the onset of Phase 2.

### D.4 THEORETICAL ERROR ANALYSIS FOR LINEAR RREGRESSION

We verify that the error in estimating the regression coefficients $a$ and $b$ in the model
$$\log(p_t) = at + b + \epsilon_t$$
scales as
$$\text{Error} \approx \text{Error}_0 \times \left(\sum_{t=1}^{k} N_t\right)^{-1/2}$$
where $N_t$ is the number of samples for $x = t$, and $\text{Error}_0$ is a constant encapsulating design matrix properties and variance factors.

#### D.4.1 PROBLEM SETUP

For each $t \in \{1, 2, ..., k\}$, we sample $N_t$ times. Each sample yields $y_{ti} \in \{0, 1\}$, and the average accuracy is computed as:
$$\hat{p}_t = \frac{1}{N_t} \sum_{i=1}^{N_t} y_{ti}.$$
Assuming large $N_t$, $\hat{p}_t$ follows approximately:
$$\hat{p}_t \sim \mathcal{N}\left(p_t, \frac{p_t(1-p_t)}{N_t}\right),$$
where $p_t$ is the true probability.

The regression model assumes that $\log(p_t)$ is linearly related to $t$:
$$\log(p_t) = at + b,$$
and after transformation, the response variable $Y_t = \log(\hat{p}_t)$ satisfies:
$$Y_t = \log(p_t) + \epsilon_t,$$

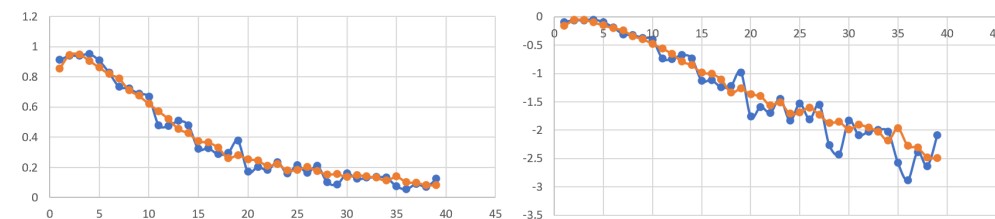

(a) Accuracy with number of operations 50 examples (blue) vs 2000 examples (coral)

(b) log(acc) with number of operations 50 examples (blue) vs 2000 examples (coral)

Figure 9: The study of the stability of a number of examples with 50 examples versus 2000 examples, we can see that variance exists but is relatively stable and small.

where

$$\epsilon_t \sim \mathcal{N}\left(0, \frac{1-p_t}{p_t N_t}\right).$$

### D.4.2 DISTRIBUTION OF $\log \hat{p}_t$

Using the Delta Method for the transformation $Y_t = g(\hat{p}_t) = \log(\hat{p}_t)$, with $g'(p_t) = \frac{1}{p_t}$, we have:

$$Y_t \sim \mathcal{N}\left(\log(p_t), \frac{p_t(1-p_t)}{N_t p_t^2}\right) = \mathcal{N}\left(\log(p_t), \frac{1-p_t}{p_t N_t}\right).$$

Thus, the transformed response variable $Y_t = \log(\hat{p}_t)$ has mean $\log(p_t)$ and variance $\frac{1-p_t}{p_t N_t}$.

### D.4.3 ORDINARY LEAST SQUARES (OLS) ESTIMATES

The OLS regression model is:

$$Y_t = at + b + \epsilon_t,$$

where $\epsilon_t \sim \mathcal{N}\left(0, \frac{1-p_t}{p_t N_t}\right)$. Let:

$$S_t = \sum_{t=1}^{k} t, \quad S_{tt} = \sum_{t=1}^{k} t^2, \quad S_p = \sum_{t=1}^{k} Y_t, \quad S_{tp} = \sum_{t=1}^{k} t Y_t.$$

The OLS estimates for $a$ and $b$ are computed by solving the Normal Equations:

$$\begin{cases} \sum_{t=1}^{k} t Y_t = a \sum_{t=1}^{k} t^2 + b \sum_{t=1}^{k} t \\ \sum_{t=1}^{k} Y_t = a \sum_{t=1}^{k} t + bk \end{cases}.$$

The solutions are:

$$\hat{a} = \frac{S_{tp} - \frac{S_p S_t}{k}}{S_{tt} - \frac{S_t^2}{k}}, \quad \hat{b} = \frac{S_p}{k} - \hat{a}\frac{S_t}{k}.$$

### D.4.4 STABILITY OF FITTED CURVE WITH RESPECT TO MORE EXAMPLES EVALUATION

Here we show what happens to the accuracy with the number of operations relationships if we increase from 50 examples per data point to 2000 examples. We select Qwen-2.5-7B-Instruct as the model of interest. The result is presented in Figure 9. We can see that the overall curve hasn't changed much. In fact, if we calculate the area under the curve for (a), it only changes from 1470 (with 50 examples) to 1490 (with 2000 examples): 1.3% difference. The result showcases that the pattern is stable very quickly.

### D.4.5 THEORETICAL ERROR ANALYSIS

The variances of the OLS estimates depend on the covariance structure of the errors. In linear regression, the covariance matrix of the estimates is:

$$\text{Cov}(\hat{\beta}) = \sigma^2 (X^T X)^{-1},$$

where $X$ is the design matrix and $\sigma^2$ is the error variance. For simple linear regression, the variances are:

$$\text{Var}(\hat{a}) = \sigma^2 \frac{1}{S_{tt} - \frac{S_t^2}{k}}, \quad \text{Var}(\hat{b}) = \sigma^2 \left( \frac{1}{k} + \frac{\bar{t}^2}{S_{tt} - \frac{S_t^2}{k}} \right),$$

where $\bar{t} = \frac{S_t}{k}$.

### D.4.6 VARIANCE OF $\epsilon_t$

The variance of $\epsilon_t$ is $\sigma_t^2 = \frac{1-p_t}{p_t N_t}$. To simplify the analysis, assume approximate homoscedasticity:

$$\sigma_t^2 \approx \frac{1-\bar{p}}{\bar{p}\bar{N}}, \quad \text{where } \bar{p} = \frac{1}{k}\sum_{t=1}^{k} p_t, \bar{N} = \frac{\sum_{t=1}^{k} N_t}{k}.$$

Define the total sample size as $\mathcal{N} = \sum_{t=1}^{k} N_t$. Since $\bar{N} = \mathcal{N}/k$, the effective error variance scales as:

$$\sigma^2 \propto \frac{1}{\mathcal{N}}.$$

### D.4.7 SCALING OF ERRORS WITH $\mathcal{N}$

Using the approximation $\sigma^2 \propto \frac{1}{\mathcal{N}}$, the variances of the OLS estimates become:

$$\text{Var}(\hat{a}) \propto \frac{1}{\mathcal{N}}, \quad \text{Var}(\hat{b}) \propto \frac{1}{\mathcal{N}}.$$

Taking square roots to obtain standard errors:

$$\text{SE}(\hat{a}) \propto \frac{1}{\sqrt{\mathcal{N}}}, \quad \text{SE}(\hat{b}) \propto \frac{1}{\sqrt{\mathcal{N}}}.$$

### D.4.8 FINAL ERROR EXPRESSION

The final expression for the regression error is:

$$\text{Error} \approx \text{Error}_0 \times \left( \sum_{t=1}^{k} N_t \right)^{-1/2},$$

where $\text{Error}_0$ encapsulates variability due to $p_t$ and the properties of the design matrix (e.g., $S_{tt} - \frac{S_t^2}{k}$). This confirms the scaling of estimation error with the total sample size.

### D.4.9 CONFIDENCE INTERVAL CALCULATION

We calculate confidence intervals for the regression coefficients using their standard errors and the inverse error function ($\text{erf}^{-1}$):

$$\text{CI} = \text{Coefficient} \pm \left( \text{SE} \times \sqrt{2} \times \text{erf}^{-1}\left( \frac{C}{100} \right) \right)$$

where $C$ is the confidence level (e.g., 95%).

### D.5 SUMMARY

Our data analysis and regression methodology accurately capture the relationship between task complexity and model accuracy during the exponential decay phase. By carefully selecting data ranges, we ensure reliable regression results that provide valuable insights into models' abilities to handle complex reasoning tasks over extended contexts.

## E FACTORS AFFECTING THE METRICS

### E.1 PRETRAINING STRATEGIES

We investigate how different pretraining strategies influence the Complexity Decay Factor (CDF), Contextual Decay Offset (CDO), and especially the Effective Complexity ($N_{\text{eff}}$) in the FACTOR benchmark. By training models from scratch using various methodologies, we aim to understand their effects on the models' abilities

to handle task complexity and context length. See Appendix F for training settings and Appendix G for training data distribution. Models were trained using the following strategies:

(1) **Baseline**: Trained directly on the FACTOR benchmark training set (regenerated to avoid data leakage), mirroring the benchmark's organization. The max context length is limited to 1000 tokens. (2) **Naive Packed Training** (*Packed*): Sequences in a mini-batch are concatenated to form longer training inputs. (3) **Question-Masked Training** (*Masked*): The question portion (payload) is masked during loss computation. (4) **Clean Context Training** (*Clean*): The model is exposed only to the payloads, without filler text. (5) **Packed Training with Diagonal Attention Mask** (*Diagonal*): Similar to Packed, but with diagonal attention masks to prevent cross-sequence attention; position IDs are retained.

We evaluated the models on the FACTOR benchmark with zero filler context to assess their baseline performance. The results are presented in Table 11.

Table 11: Effects of Pretraining Strategies on Model Performance (Zero Filler Context).

| Model | CDF | CDO | $N_{\text{eff}}$ |
|---|---|---|---|
| Baseline | $-0.3292$ | 8.4685 | 25.72 |
| Packed | $-0.1339$ | 2.6381 | 19.70 |
| Masked | $-0.3278$ | 8.5196 | 25.99 |
| Clean | $-0.2672$ | 7.0388 | 26.34 |
| Diagonal | $-0.3780$ | 10.0887 | 26.69 |

The Effective Complexity ($N_{\text{eff}}$) indicates the maximum task complexity the model can handle before performance significantly degrades. Higher $N_{\text{eff}}$ values reflect better performance over a broader range of complexities.

From Table 11, we observe: (1) The **Baseline**, **Masked**, **Clean**, and **Diagonal** models achieve similar $N_{\text{eff}}$ values around 26, indicating they handle task complexities up to $N \approx 26$ effectively. (2) The **Packed** model has a lower $N_{\text{eff}}$ of 19.70, suggesting worse performance over the range of task complexities. Naive packed training leads to poorer handling of complex tasks compared to other strategies. Despite a more negative CDF ($-0.3780$), the **Diagonal** model achieves the highest $N_{\text{eff}}$ (26.69) and a higher CDO (10.0887). This means the model starts with higher baseline accuracy (due to higher CDO) and maintains decent performance for $N$ between 25 and 33, extending the effective complexity range. **Packed Training** reduces $N_{\text{eff}}$, leading to worse performance across task complexities. **Diagonal Training**, although it has a more negative CDF, extends $N_{\text{eff}}$, improving performance for some higher $N$ values. Therefore, while naive packed training negatively impacts the model's ability to handle complex tasks, the Diagonal strategy enhances performance at higher complexities, evidencing its benefit in extending the range over which the model maintains accuracy.

### E.2   POST-TRAINING STRATEGIES

We examine how different fine-tuning strategies affect model performance on the FACTOR benchmark, particularly the CDF and CDO metrics, with a maximum context length of 16K tokens. Notably, although the maximum training length is 16K, the majority of sequences in the training distribution are less than 8K. The models are trained based on the checkpoing of the Baseline model in the last section. We evaluate three approaches: (1) **Gradual Sequence Length Increase** (*Course*): Starting with shorter sequences and progressively increasing the length during fine-tuning. (2) **Mixed-Length Sequence Training** (*Mixed*): Training on sequences of varied lengths simultaneously. (3) **Direct Long Sequence Training** (*Full*): Fine-tuning directly on the maximum sequence length without gradual adaptation.

Table 12 and 13 present the CDF and CDO metrics at different context lengths.

The fine-tuning strategies exhibit distinct effects on CDF and CDO:

Despite training up to 16K contexts, the *Course* and *Mixed* strategies significantly outperform the *Full* strategy, likely due to the training distribution containing mostly shorter sequences. Sudden exposure to full-length sequences in the *Full* strategy causes a distribution shift, leading to degraded performance.

**Course** strategy results in models with stable CDO and degrading CDF, resembling Llama series models. This indicates that the models maintain baseline performance but struggle more with task complexity as context length increases. **Mixed** strategy yields models with stable CDF and degrading CDO, similar to

Table 12: CDF at Different Context Lengths. (Values scaled by $10^{-2}$)

| Model | 0 | 1K | 2K | 4K | 8K | 16K |
|-------|-----|-----|-----|-----|-----|-----|
| *Course* | $-7.91$ | $-6.44$ | $-8.28$ | $-8.31$ | $-10.29$ | $-14.56$ |
| *Mixed* | $-8.43$ | $-6.66$ | $-7.79$ | $-6.86$ | $-8.11$ | $-11.25$ |
| *Full* | $-12.48$ | $-9.34$ | $-17.28$ | $-12.42$ | $-5.76$ | $-18.82$ |

Table 13: CDO at Different Context Lengths.

| Model | 0 | 1K | 2K | 4K | 8K | 16K |
|-------|-----|-----|-----|-----|-----|-----|
| *Course* | 0.4775 | 0.3539 | 0.4916 | 0.3383 | 0.3392 | 0.1038 |
| *Mixed* | 0.3227 | 0.1351 | 0.2076 | 0.0080 | $-0.0922$ | $-0.3469$ |
| *Full* | $-0.3222$ | $-0.4789$ | $-0.2658$ | $-0.3260$ | $-0.7427$ | $-0.2435$ |

some pretrained models that handle task complexity consistently but whose baseline performance declines with longer contexts, resembling models like gpt-4o-mini. **Full** strategy leads to poor performance in both metrics, due to abrupt changes in data distribution. Gradual adaptation to longer sequences during fine-tuning helps models cope better with increasing context lengths and task complexities, mitigating failure modes observed in pretrained models.

### E.3 REPEATED SAMPLING STRATEGIES

We analyze how increasing computational efforts during inference affects model performance, particularly focusing on the accuracy and the effective complexity $N_{\text{eff}}$. To investigate the impact of increased computational effort during inference, we employ **repeated sampling**. This approach measures the probability of correctly solving at least one instance out of $t$ samples. We conduct experiments on models trained with the Gradual Increase strategy at different filler context lengths, as well as on the model pre-trained with a clean context (zero filler context).

Table 14 summarizes the linear regression results for different filler context lengths. The regression equation is given by:

$$\log(N_{\text{eff}}^{\text{clean}} - N_{\text{eff}}^{\text{model}}) = k\log(t) + b$$

Table 14: Linear Regression Results for Different Filler Context Length.

| Filler Context Length | $k$ (Slope) | $b$ (Intercept) | R-squared |
|-------|-----|-----|-----|
| 0 | -0.358287 | 2.865435 | 0.992356 |
| 1000 | -0.333536 | 2.670190 | 0.987057 |
| 2000 | -0.376972 | 2.865891 | 0.982553 |
| 4000 | -0.261895 | 2.999738 | 0.995347 |
| 8000 | -0.245732 | 3.242366 | 0.992648 |
| 16000 | -0.150702 | 3.524461 | 0.998234 |

Figure 10 illustrates the relationship between $N_{\text{eff}}$ and the number of tries $t$.

Key observations from the results are: (1) For models fine-tuned with gradually increasing context lengths, increasing the number of samples $t$ leads to a slow improvement in $N_{\text{eff}}$, but it remains below the $N_{\text{eff}}$ of the clean context model. (2) **Effect of Context Length**: As the filler context length increases, the value of $k$ becomes less negative, indicating a decrease in the rate of change. Simultaneously, the intercept $b$ increases. (3) **Interpretation**: While increasing computational efforts during inference provides marginal benefits, it does not fully overcome the challenges posed by high task complexity and long contexts. There exists spontaneous bias within the model. For higher complexity tasks, the benefit of repeated sampling is diminishing, revealing that for certain questions, achieving the correct answer though repeated sampling is nearly impossible. This raises concerns about the model's generalization performance with respect to out-of-training-distribution complexity.

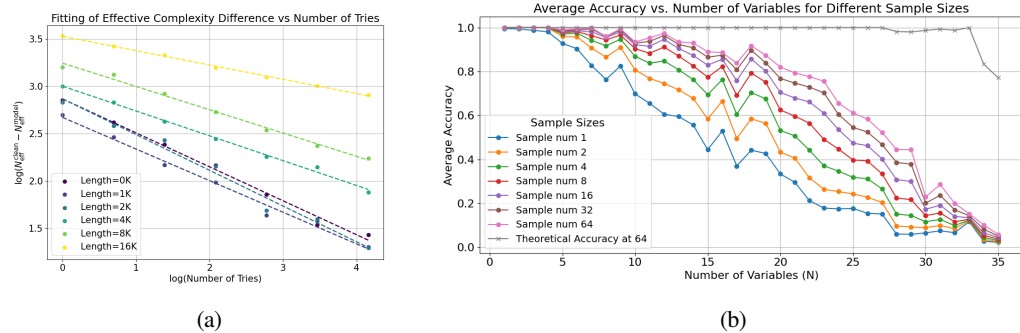

(a) (b)

Figure 10: (a) exhibits the strong linear correlation between $\log(N_{\text{eff}}^{\text{clean}} - N_{\text{eff}}^{\text{model}})$ and $\log(t)$. This implies a slow improvement in $N_{\text{eff}}$ with the increase of number of samples, and it does not fully overcome the challenges posed by high task complexity and long contexts. (b) is the average accuracy as a function of the number of variables (N) for different sample sizes. The plot reveals that For higher complexity tasks, the benefit of repeated sampling is diminishing.

## F  TRAINING CONFIGURATIONS

In this section, we provide a concise overview of the training configurations used for developing our language models evaluated on the FACTOR benchmark. Key configurations are summarized in Table 15, and additional details are described to clarify the training setup.

Table 15: Summary of Training Configurations.

| Configuration Element | Setting |
|---|---|
| Model Architecture | LlamaForCausalLM |
| Number of Layers | 12 |
| Number of Attention Heads | 12 |
| Hidden Size | 768 |
| Intermediate Size | 3,072 |
| Vocabulary Size | 424 |
| Maximum Position Embeddings | 32,768 |
| RoPE Theta | 500,000.0 |
| Data Type | `bfloat16` |
| Peak Learning Rate | 4e-4 |
| Batch Size | 192 |
| Learning Rate Scheduler | Cosine decay with warmup ratio 0.01 |
| Number of Training Epochs | 1 |
| Optimizer | AdamW |

### F.1  MODEL ARCHITECTURE

We employ a Llama3 architecture configured to handle long sequences and complex reasoning tasks. The model consists of 12 Transformer layers, each with 12 attention heads, resulting in a hidden size of 768 (calculated as `num_heads` $\times$ 64). The intermediate size is set to 3,072 (four times the hidden size), following common practice to enhance model capacity.

To effectively handle long contexts, we use Rotary Position Embeddings with a RoPE theta value of 500,000.0. This allows the model to capture positional information over extended sequences. We utilize FlashAttention-2 for efficient attention computation on long sequences, which improves training speed and reduces memory consumption.

All computations are performed using `bfloat16` precision to optimize memory usage and computational efficiency without significantly affecting model accuracy.

### F.2 TOKENIZER

A custom tokenizer is used, tailored to the specific needs of the FACTOR benchmark tasks. The tokenizer has a vocabulary size of 424 tokens.

**Optimizer and Scheduler** The AdamW optimizer is utilized with default parameters except for the learning rate. The cosine learning rate scheduler with warmup helps in gradually adapting the learning rate, reducing the risk of training divergence at the start.

## G TRAINING DATA GENERATION

This section details the generation of the training data used in both the pretraining and posttraining phases. Our approach carefully controls task complexity and context length to effectively train the models for evaluating their reasoning abilities on the FACTOR benchmark.

### G.1 PRETRAINING DATASET

The pretraining dataset comprises 3 million synthetic Question-Solution pairs. It is designed to teach the model fundamental reasoning skills over a range of task complexities and shorter context lengths.

#### G.1.1 TASK COMPLEXITY DISTRIBUTION

To control task complexity, we vary the number of variables $N$ in each synthetic example. The value of $N$ is determined by:

$$N = \min(N_1, N_2), \quad N_1, N_2 \sim \text{Uniform}(1, 29)$$

where $\text{Uniform}(1, 29)$ denotes a discrete uniform distribution over integers from 1 to 29 (excluding 30). By taking the minimum of two independently sampled values, we bias the distribution toward smaller values of $N$, emphasizing simpler tasks while still including examples with higher complexities up to $N = 29$.

#### G.1.2 FILLER CONTEXT LENGTH DISTRIBUTION

The length of the filler context $L_{\text{pretrain}}$ is sampled from a log-normal distribution to introduce variability in context lengths while maintaining manageable sequence sizes for pretraining. Specifically, we use:

$$\ln L \sim \mathcal{N}(\mu_{\ln L}, \sigma_{\ln L}^2), \quad L_{\text{pretrain}} = \exp(\ln L) - L_{\min}$$

where: $\mu_{\ln L} = 5$, $\sigma_{\ln L} = 2.5$, $L_{\min} = 100$, $L_{\max} = 1000$

After sampling $\ln L$, we compute $L$, round it to the nearest integer, and clip it to the range $[L_{\min}, L_{\max}]$. We then subtract $L_{\min}$ to adjust the lengths to start from zero. This results in a distribution of filler context lengths biased toward shorter lengths but including a range up to 900 tokens.

### G.2 POST-TRAINING DATASET

The post-training dataset consists of 60,000 synthetic long-context Question-Solution pairs. It is used to fine-tune the pretrained model, enhancing its ability to handle extended contexts and complex reasoning tasks.

We employ three different training strategies to investigate their effects on model performance:

1. **Gradually Increasing Context Length**: The dataset is divided into four equal parts, each containing 15,000 examples. Each part corresponds to filler context lengths that are progressively increased. Specifically, the filler context length distribution in each stage is scaled by factors of 2, 4, 8, and 16 times the pretraining filler context length $L_{\text{pretrain}}$, respectively. By gradually increasing the context length, the model is incrementally adapted to handle longer sequences, reaching the maximum context length in the final stage.

2. **Mixture of Different Context Lengths**: In this strategy, the examples from all four stages of the gradually increasing context length dataset are combined and shuffled. Training on this mixed dataset exposes the model to different context lengths simultaneously, which may encourage better generalization across various sequence lengths.

3. **Training Directly with Full Context Length**: The model is trained starting from a filler context length distribution scaled by $16 \times L_{\text{pretrain}}$. This approach tests the model's ability to handle long contexts without prior adaptation through shorter sequences.

### G.2.1 TASK COMPLEXITY DISTRIBUTION

For all posttraining strategies, the number of variables $N$ is sampled using the same method as in pretraining:

$$N = \min(N_1, N_2), \quad N_1, N_2 \sim \text{Uniform}(1, 29)$$

This ensures consistency in task complexity across both pretraining and posttraining datasets.

### G.2.2 FILLER CONTEXT LENGTH GENERATION

For each stage in the gradually increasing context length strategy, the filler context lengths are generated by scaling the pretraining filler context lengths $L_{\text{pretrain}}$:

$$L_{\text{stage}} = s \times L_{\text{pretrain}}$$

where $s$ is the scaling factor for each stage ($s = 2, 4, 8, 16$).

The filler context lengths are generated using the same log-normal distribution as in pretraining but adjusted for the scaling factor. This adjustment ensures that the filler context lengths are appropriately scaled for each stage.

### G.2.3 SYNTHETIC DATA GENERATION PROCESS

For each example in both the pretraining and posttraining datasets, we follow the same synthetic data generation steps as described in Section C, adjusting the filler context lengths according to the stage and strategy.

### G.3 SUMMARY

By carefully generating training data with controlled task complexity and systematically varied context lengths, we aim to investigate the impact of different training strategies on the models' abilities to handle complex reasoning tasks over long contexts. The datasets are designed to provide comprehensive exposure to the challenges posed by increased sequence lengths and to facilitate a detailed analysis of model performance under various training conditions.

