# OpenReview forum: "FACTOR: Factoring Complexity and Context Length in Long-Context Model Evaluation"
_ICLR.cc/2025/Conference — Submitted to ICLR 2025_

### Official Review · Reviewer_zsAb · 2024-10-29

**Soundness:** 4
**Presentation:** 4
**Contribution:** 4
**Rating:** 8
**Confidence:** 4

**Summary:**

This paper proposes FACTOR - a novel benchmark for evaluaition of long context reasoning in LLMs. Most importantly - it separates task complexity from context length. Then the paper evaulates several LLMs on this benchmark and shows that performance has a log linear relationship with complexity. Two metrics were then derived from this relationship - CDF and CDO. This is important to quantify the degradation in reasoning with complexity and length. Then the paper tests impact of pretraining and finetuning on these metrics, as well as the impact of more frequent sampling during inference.

**Strengths:**

Originality - I believe that CDO and CDF are great metrics that fill a gap in LLM literature, and the FACTOR benchmark is very useful for this.

Quality - The methodology makes sense and the experiments are well structured. I found the log linear result to be very interesting.

Clarity - the paper is well written, easy to follow.

Significance - These results are an important addition to scaling laws, and important findings.

**Weaknesses:**

Not much to say but -

1. Task diversity is limited, which means maybe this is not as representative of real world scenarios.
2. FACTOR seems to be a very computationally expensive benchmark to run.

**Questions:**

1. Would FACTOR be extended to further logical inference/ commonsense reasoning tasks?
2. How robust are CDO and CDF to randomness of seeds used in FACTOR generation? Are there any analyses on stability of these metrics?

---

> ### Author Response · Authors · 2024-11-24
> **Response to zsAB**
>
> Thank you so much for your attention to our paper. We really appreciate that you think of our benchmark originality and acknowledge the quality of our evaluation. We will try our best to address your concerns about dthe iversity of the dataset, computational cost, stability of the metrics, and others. If you feel that your concerns have been addressed, please consider raising your score.
>
>  - Task diversity is limited
>
> Thank you for such an insightful comment. To increase the diversity of our current dataset, we introduce an new problem generation system for generating common-sense reasoning problems of large volume with fine-grained control. (More details about the common sense problem generation in $\text{\textcolor{blue}{Section 4.2, page 6}}$) To summarize, we see the potential in mapping between computational graph and natural language problems from the old FACTOR and we apply its spirit onto real-world context. The result is the new FACTOR in the revised paper, containing three subset of tasks classified based on hierarchical complexity in natural language structure (elaborated intensively in paper). The easy subset contains the old FACTOR, while the medium and hard subsets contains the reasoning problem in the real-world context. The latter two contains problems similar to the ones in GSM-8K and has 19 splits of problems, ranging from #ops = 2 to #ops = 19). To further increase the diversity of the tasks, we use three different real-world context template, together with both forward reasoning and backward reasoning modes. More details and evaluations can be read in the revised paper. Benefitting from this move, we can evaluate LLM on contextual problems that can be easily scaled up in difficulty. Below, we show results of Llama-3.1-70B-Instruct evaluation on these three subsets. You can clearly see that the rate for the accuracy degradation is very different, Easy subset has the slowest decay rate, while Hard subset decay the fastest.
>
> | op | Easy | Medium | Hard |
> |-----|--------|--------|--------|
> | 5 | 0.808 | 0.86 | 0.7 |
> | 8 | 0.832 | 0.875 | 0.512 |
> | 10 | 0.708 | 0.612 | 0.41 |
> | 12 | 0.628 | 0.56 | 0.36 |
> | 15 | 0.556 | 0.43 | 0.19 |
> | 20 | 0.516 | 0.2 | 0.05 |
>
>  - FACTOR is computationally expensive?
>
> We appreciate the concern about computational cost when running the full benchmark. We admit that running the full benchmarks is computationally expensive. However, our proposed results modeling is here to help. We found that the exponential decaying trend actually fit the measured experiments quite well. Therefore, it provides you an opportunity to only subsample some of the ops from 1 to 39, then run an evaluation on them then extrapolate onwards on missing op between 1 and onwards seems to provide a reasonable solution. Below we subsample from the FACTOR EASY subtask using 8B and 70B. You can see that as we increase the subsample ratio or the stride, the MSE of the fitted log-linear curve starts to fluctuate. But mostly doesn’t differ from an order of magnitude. It seems that strides 3 and 4 are reasonably close to the baseline here which is stride equaling 1, or no subsampling. Therefore, the computation can be shrunk by ~3x - 4x with a reasonable loss in precision.
>
> | Stride | Llama-3.1-8B-Instruct Easy | Llama-3.1-70B-Instruct Easy |
> |--------|----------------------------|-----------------------------|
> | 1 | 0.003196 | 0.002698 |
> | 2 | 0.004627 | 0.003270 |
> | 3 | 0.005541 | 0.003662 |
> | 4 | 0.005438 | 0.002643 |
> | 5 | 0.005645 | 0.002822 |
> | 6 | 0.008988 | 0.005842 |
> | 7 | 0.004291 | 0.002979 |
> | 8 | 0.003345 | 0.003906 |
>
> Also, one other way is to test on the Hard subset where the 8B model hardly gives reasonable results when op > 5, compared to EASY when 8B has a score at 20% accuracy even when op = 20, which would drastically shrink the number of ops to run to grasp the full landscape of model performance.
>
> An interesting note when we approach this problem, we discover a pleasing insight about the error of the fitted CDO and CDF, that is when assuming the decay of accuracy is exponential with respect to the number of operations, the error is proportional to the inverse of square root over the total number of examples measured. We found that this law doesn’t require all ops need be measured, it is only related to the total number of examples you measured. More elaboration and proofs are in the $\text{\textcolor{blue}{Appendix D4.1 - D4.3 and D4.5, pages 22 - 23}}$ of the paper.

---

> ### Author Response · Authors · 2024-11-24
> **Continue the response to zsAb**
>
> - Stability of CDO and CDF of the fitted curve when changing seeds
>
> Thank you for the insightful question. We found that the result is very robust and stable. Below is the study that would cover the case you are specifically concerned about. We contrast between two different settings, for FACTOR EASY, we select the Qwen 2.5 7B model and run it for the entire 1 to 39 subsets. The difference is that for one scenario, we evaluate only 50 examples for every value, while for the other scenario, we evaluate 2000 examples for every value. (You can think of 2000 as we run the first case 40 times but each with a different seed). We compare the area under the curve for these two, for 50 it is 1470, while for 2000, it is 1490, differing by 1.3%. The gap is minimal. In the $\text{\textcolor{blue}{Appendix D4.4, page 23}}$ of the paper, we also present the figure of this experiment, we can see that the 50 example case has roughly the same shape as the 2000 version as well.

---

> > ### Author Response · Authors · 2024-11-27
> > **Response to zsAb**
> >
> > We appreciate your attention to our work and your inspiring feedback! It is approaching the paper edition deadline. If you feel that any critical experiments or results are missing, please let us know immediately so we can try our best presenting them in the revised paper to dispel your concerns and clarify your confusions with additional soundness. Thank you so much.

---

### Official Review · Reviewer_ALNr · 2024-11-03

**Soundness:** 2
**Presentation:** 3
**Contribution:** 2
**Rating:** 3
**Confidence:** 3

**Summary:**

This paper performs an experimental analysis of several language models for a particular reasoning task solving sets of simple equations when context windows increase and separate those equations. The results are fitted with simple linear equations (after logarithmic scale transformations) and two indicators are derived from the linear functions (the slope, Complexity Decay Factor, and the y-intercept, the Contextual Decay Offset). Some insights are extracted from that modelling.

**Strengths:**

The paper is well written, and it is easy to follow.
There is more need to show how context windows affect some reasoning process and no appropriate benchmarks seem to exists, and previous analyses are confounded by other factors. So there's motivation for this paper.
Many models, including very recent ones, are covered. The experiments are quite exhaustive in that dimension.

**Weaknesses:**

The kind of task is relatively narrow, and different results could be observed for other tasks. In particular, the task question is formulated as choosing those variables that meet the requirements. This can have elimination/random components that are specific to this task and do not generalise to other tasks, or even some other presentations of this task (asking for actual values, instead of affected variables). There's not sufficient analysis of shortcuts, even if the authors claim they ensure they not happen. The paths to the right answers may be short in some cases and this is intrinsic with the problem and not random.

Apart from the use of only one problem, the major problem is the fitting. The authors calculate the logarithm of the y-axis on a bounded performance scale that goes between 0 and 1. But this can never give a linear relationship if we extend the complexity on the x-axis until we reach 0 accuracy (more variables or questions where choices or elimination make success by pseudochance very unlikely). The full response curves are actually sigmoid and the observed behaviour in this paper happens because collected data stops in a high-slope part of the sigmoid. Given the character of the y-axis, a logistic fit on the original scale would have made much more sense than a linear fit on a logarithmic scale.

In fact, extracting Effective Complexity using the linear model has the problem of possible extrapolation of a negative number of variables, something that wouldn't happen with a logistic model. With a logistic model, the effective complexity could simply be chosen when accuracy goes below a threshold of 90%.

The comparison with RAG appears in the abstract and the introduction, but then this disappears in the paper, so to me it's thrown as a red herring and deviates the motivation.

More justification is needed for repeated sampling as a proxy for increasing inference compute. Actually, there's no need to do experiments for this, as the tries are independent and it's a question of multiplying the failure probability of each of it. That's why figure 6 looks so linear.


Minor details:
- acronym in the abstract wrong
- the related work section has some expression and formatting issues (blank spaces missing before the references).

**Questions:**

I couldn't find how scoring of answers is performed. For instance, if the ground truth is v3 and v5, what's the score if the model says only v3? Is it only totally right or otherwise wrong? This is very important to understand whether there are shortcuts or not.

About o1-mini, it is said that "inference-time strategies can significantly enhance performance across task complexities", but we don't know (or at least no source is provided) of whether o1 is using some other techniques that affect the context window, or it gives several passes or other strategies, since the details of the system have not been published. Do you have more details about o1 that are publicly available?

Can you explain why repeated sampling is a proxy for increasing inference compute?

Can you make a logistic fitting instead of a linear fitting?

Do you think the results will hold for other tasks?

Where's the discussion/comparison about RAG?

---

> ### Author Response · Authors · 2024-11-24
> **Official Response to ALNr**
>
> Thank you so much for your attention to our paper. We appreciate your acknowledgment of the comprehensiveness of our evaluation. We will try our best to address your concern about the diversity of the dataset, modeling the results, RAG, and others. If you feel that your concerns have been addressed, please consider raising your score.
>
>  - Lack of diversity of the benchmark and also miscommunication of the benchmark grading
>
> In the new version of FACTOR, we address the mentioned narrowness from the old benchmark.
> To increase the diversity of our current dataset, we introduce a new problem-generation system for generating common-sense reasoning problems of large volumes with fine-grained control. (More details about the common sense problem generation in $\text{\textcolor{blue}{Section 4, page 6 - 8}}$) To summarize, we see the potential in mapping between computational graph and natural language problems from the old FACTOR and we apply its spirit to real-world context. The result is the new FACTOR, containing three subsets of tasks classified based on hierarchical complexity in natural language structure (elaborated intensively in the revised paper). The easy subset contains the old FACTOR, while the medium and hard subsets contain the reasoning problem in the real-world context. The latter two contain problems similar to the ones in GSM-8K and have 19 splits of problems, ranging from #ops = 2 to #ops = 19). To further increase the diversity of the tasks, we use three different real-world context templates, together with both forward reasoning and backward reasoning modes. Below, we show the results of Llama-3.1-70B-Instruct evaluation on all three subsets. The easy subset has the slowest decay rate, while the Hard subset decays the fastest.
>
> | op | Easy | Medium | Hard |
> |-----|--------|--------|--------|
> | 5 | 0.808 | 0.86 | 0.7 |
> | 8 | 0.832 | 0.875 | 0.512 |
> | 10 | 0.708 | 0.612 | 0.41 |
> | 12 | 0.628 | 0.56 | 0.36 |
> | 15 | 0.556 | 0.43 | 0.19 |
> | 20 | 0.516 | 0.2 | 0.05 |
>
> Also, some clarification of the later questions about problem design and parsing implementation. For the Easy subset (the old FACTOR), we ask about all variables that have a certain value. We don’t have partial credit for questions: it either is correct because all variables are found out correctly or not correct. Therefore, no shortcut can be made in the general case, the model has to go through the entire computation graph and go for the exact number of operations listed. Remember that the problem is based on a randomly generated computation graph, while the variables to begin solving the graph have randomly generated values. Therefore, to consistently solve the problem correctly, the entire computation graph has to be traversed. The graph structure is random with no priors.
>
> For the newly added common sense questions in the Medium and Hard subsets, the solution generated is based on a topological sort of computation DAG randomly generated. The number of ops is determined based on the length of the topological sort of the graph. Therefore, it is impossible to find a shorter path to solving the problem.
>
>  - Logistic Regression vs. Two Phase constant plus loglinear decay
>
> We appreciate the thoughtful discussion about why you prefer logistic regression. To settle this argument, we identify three cases and fit both the log-linear and logistic regression. However, based on our observation from the studies, the decaying exponential seems to be a better option for accurately describing the trend.  Here o1-mini is on Easy subset is the only case where we are dealing with this first plateau and then decay scenario for op > 0. We show visualization in $\text{\textcolor{blue}{Figure 7, page 7 - 8}}$ of the revised paper.
>
> For FACTOR Easy with Llama-3.1-8B-Instruct and Llama-3.1-70B-Instruct
> | Model | Loglinear Fit | Logistic Regression Fit |
> |-----------------------------------------|---------------|--------------------------|
> | O1-mini on FACTOR easy | 0.01349 | 0.0062 |
> | Llama-3.1-8B-Instruct on FACTOR easy | 0.0031 | 0.0261 |
> | Llama-3.1-70B-Instruct on FACTOR-easy | 0.0026 | 0.0413 |
> | Llama-3.1-8B-Instruct on FACTOR medium | 0.0019 | 0.0033 |
> | Llama-3.1-70B-Instruct on FACTOR medium | 0.0042 | 0.0165 |
>
> Consistently, as you can see below, the log-linear always gives a much better fit (sometimes by one order of magnitude). We use MSE for metrics. The O1-mini model has this phase transition from near 1 perfect to starting to decay. Our paper uses a two-phase step-wise function consisting of a constant plus an exponential decay. It achieved a lower and better MSE score compared to the logistic regression fit. As can be seen from the visualizations, once the model starts decaying, it drops steeply, rather than smoothly at first then accelerate as in logistic regression. As for the possible negative extrapolation, we now avoid it by clearly stating the support:  the ops count for the curve has to be positive.

---

> ### Author Response · Authors · 2024-11-24
> **Continue the response to ALNr**
>
> - RAG experiments are missing
>
> Here we present thorough evaluations of studies of RAG on our dataset. Thanks to our well-conceived noise, FACTOR medium and hard effectively prevent RAG from solving the question entirely. (Details about the noise addition in $\text{\textcolor{blue}{Section 4.4, page 7}}$).
>
> | op | RAG (with Llama 3.1 8B-Instruct) >2k | Llama-3.1-8B-Instruct 8K | RAG (with Llama 3.1 70B-Instruct) >2k | Llama-3.1-70B-Instruct 8K |
> |----|--------------------------------------|---------------------------|---------------------------------------|---------------------------|
> | 2 | 0.13 | 0.802415 | 0.175 | 0.975 |
> | 3 | 0.11 | 0.532258 | 0.07 | 0.835 |
> | 4 | 0.055 | 0.679435 | 0.145 | 0.92 |
> | 5 | 0.095 | 0.264113 | 0.1 | 0.87 |
> | 6 | 0.08 | 0.233871 | 0.12 | 0.705 |
> | 7 | 0.065 | 0.280242 | 0.085 | 0.655 |
> | 8 | 0.055 | 0.245968 | 0.05 | 0.59 |
> | 9 | 0.05 | 0.169355 | 0.07 | 0.38 |
> | 10 | 0.045 | 0.137097 | 0.045 | 0.39 |
>
> The result of our high-quality noise addition is that RAG is no longer effective in solving our tasks. (Referring to revised $\text{\textcolor{blue}{Section 3 page 4 - 5}}$) for more) We build a standard RAG system using Llama 3.1 8B Instruct as the decoder while using the al-mpnet-base-v2 for the retriever. We evaluated the system on various previous and standard long context datasets to show that the RAG delivers reasonable performance. Here we highlight the Variable Tracking (VT) from RULER, our method gives 86% acc, whereas LLM directly gives 99%. We use problem context 8K while the budget for RAG is to be 2K. We tested this RAG system on our FACTOR medium subset, and we found that it quickly dropped to zero. We also replaced the decoder with Llama 3.1 70B Instruct, and we found that it is the same pattern.
>
> | | 8K Iterative Prefill RAG | 8K (Llama-3.1-8B-Instruct) |
> |---------------------|--------------------------|----------------------------|
> | VT from RULER| 100% | 99.2% |
> | 2 | 0.52 | 0.8024 |
> | 4 | 0.22 | 0.6794 |
> | 5 | 0.0729 | 0.2641 |
> | 6 | 0.02 | 0.2338 |
> | 8 | 0.06 | 0.2459 |
>
> To make the case more convincing, we now enable iterative prefilling for the RAG side. We disregard the efficiency of the RAG system and test its performing ceiling. It now reaches 100% on the VT. However, it still struggles heavily on our FACTOR medium when op > 3. You can see that the accuracy quickly drops to zero. Therefore, RAG cannot determine the essential logic path in its chunk embedding space. Please refer to the revised paper for more details.
>
>  - What is the point of repeated sampling
>
> Thanks for asking the insightful questions. We acknowledge that repeated sampling is one technique for scaling up inference time computation, but it isn’t the only way. On the other hand, many recent works (see citations below) show that repeated sampling (LMM, Aviral Kumar) leads to strong improvement in math and coding generation tasks.
>
> Brown, B., Juravsky, J., Ehrlich, R., Clark, R., Le, Q. V., Ré, C., & Mirhoseini, A. (2024). Large language monkeys: Scaling inference compute with repeated sampling. arXiv preprint arXiv:2407.21787.
>
> Snell, C., Lee, J., Xu, K., & Kumar, A. (2024). Scaling llm test-time computationally optimally can be more effective than scaling model parameters. arXiv preprint arXiv:2408.03314.
>
> From our study, we found that repeated sampling is most effective in the first half of the ops range compared to the second half out of the 1 - 39 ops range. Please refer the Figure 8(b) from the appendix. Its effectiveness diminishes for harder problems. For op 15, sampling 64 times boosts the performance from around 45% to around 90%. However, for op 30, the increment from zero to 64 only increases the performance from around 5% to 20%. We hope this finding to be helpful for the community to know when reading our benchmark.
>
>  - Scoring Miscommunication
>
> Thank you for expressing your concern. In our grading system for the FACTOR easy task, a partial answer of an incomplete list of variables is counted wrong. The score can either be 0 or 1 and only when the answer is precisely v3 and v5 does the score become 1. **No partial result is allowed.**
>
>  - Result holds for other tasks
>
> In $\text{\textcolor{blue}{Table 5, page 9}}$, you can see that our evaluation of the FACTOR Easy subset produces rankings on correlate closely with the LMSYS ELO score, which is widely accepted as the popular benchmark for evaluating the LLM generation ability.

---

> > ### Comment · Reviewer_ALNr · 2024-11-24
> >
> > Thank you for the thorough responses and extensions to the paper. I still have conceptual issues about the generality of the tasks, even if more tasks are included now, and the functions used for fitting the data. The data seems to be stopped to fit a linearlog, hence if fits the linearlog model better than the logistic. Also, the RAG analysis now makes the paper more convoluted. Overall, there is significant effort in this paper, but my concerns about the contribution and the justification for some choices remain.

---

> ### Author Response · Authors · 2024-11-25
> **Response to ALNr**
>
> Thank you for your speedy response to our posts. Also, thank you for acknowledging our effort. We summarize the two lingering concerns you have and try our best to address them. If you feel that your concern is addressed, please consider raising the score.
>
>  - The comparison between exponential and logistic regression.
>
> We appreciate your acute observation that the curve of the accuracy decay of LLM versus operation count (op) on our dataset has three different parts: a plateauing phase at the beginning when op is close to zero and the question is easy; the rapid decaying phase when op increases; and a plateauing phase for accuracy around zero when the op is high and LLM no longer capable of solving the problem. **Therefore, it is natural to use logistic regression, a function so elegant that can cater to all three phases at once, showcasing your deep observation about our tendency.** We want to assure the reviewer that we have taken this proposal seriously, and we also like this idea a lot. In fact, we have **modified** the analysis portion in $\text{\textcolor{blue}{Section 5.1, page 8}}$ to take this into our analysis. In contrast, we proposed to use **two functions** (a step-wise function) to explain what we observed from the curve. (In the paper, we write it as two-phase behavior in Section 5.1) First, a constant function to explain the initial plateauing stage, followed by an exponential decay function to explain the subsequent two stages. In the paper, we write the second stage as "log linear", so fitting log(y) = a'x + b'. Thus, y = exp(a'x + b'), a' < 0 since it is decaying. It can be written as 1/exp(ax + b) for a positive number a and b.
>
> The two approaches are almost identical when x is positive and large since given a linear relationship of x, f(x) = ax + b. Logistic regression gives 1/(1 + exp(f(x))). When x is large, it is almost 1/exp(f(x)): this means that if a sigmoid curve can fit raw data well on the high magnitude positive regime, the raw data can also be fitted well by an exponentially decaying function. The only difference is at the transition between the initial plateau and subsequent decay, where logistic regression predicts it to be a smooth drop, where our stepwise function shows a sudden drop.  **Currently, as we have shown, the vast majority of the models barely have the initial plateauing phase due to the lack of reasoning ability.** Therefore, the focus of today is on the decaying stage, where these two functions are almost identical. Only a very handful of models such as `o1-mini` are showing all three phases. On o1-mini, the step-wise approach is shown to better fit the data than logistic regression. **When focusing on the decaying phase of the trend, we present a costly experiment running Gemini-1.5-Flash with high precision from op 1 to 160. We show the results in $\text{\textcolor{blue}{Figure 8, page 19}}$. According to the MSE score, exponential decay is better than logistic regression fit.** However, in the short future, with the advent of a plethora of stronger reasoning models, some models might have different trends, and we will update our modeling function once the trend differs. Thank you again for proposing the logistic regression model and your acute observation.
>
>  - RAG
>
> Thanks for pointing out your confusion. RAG experiments are the motivation of our benchmark. RAG experiments reveal the deficiency of the current long context benchmark: the current benchmark shows that the gap between the highly expensive to train and deploy long context LLM and the simple-to-build RAG counterparts is very close. The tremendous cost of building long-context LLM cannot be justified merely based on the current long-context benchmarks. Therefore, we need a new benchmark that truly differentiates these two, and therefore accentuates long-context LLM's value.
>
> Specifically, to prove the current long context benchmark's inefficiencies in the paper, we first show that conventional RAG is strong on previous standard long context benchmarks that are supposed to be used to evaluate LLMs. However, because of RAG's strong performance, the advantage of LLM over RAG on long context jobs is marginal and questioning. Then, we show that our tasks, because of the reasoning nature, completely prevent RAG from doing it effectively. Even if we enable iterative prefilling, which is close to the upper bound of RAG, RAG performance continues to be ineffective. In comparison, LLMs' have superiority and a huge margin over RAG on our tasks. Does that help clarify your confusion about RAG experiments' results? Let us know whether you have more lingering concerns about RAG.

---

### Official Review · Reviewer_ZEhP · 2024-11-04

**Soundness:** 2
**Presentation:** 3
**Contribution:** 2
**Rating:** 6
**Confidence:** 4

**Summary:**

The paper presents FACTOR, a benchmark specifically developed to assess the performance of LLMs in handling complex reasoning tasks over long contexts. FACTOR generates synthetic problems that require models to understand and solve relationships between variables within long pieces of text. The authors also evaluate several models by evaluating them on varying tasks complexity and context length.

**Strengths:**

They propose a task generation framework that can be used to generate tasks with various levels of difficulty and context lengths.

They evaluate multiple models using their benchmarks, and show that by increasing the context length and difficulty of the problems, these models struggle to reason correctly on them.

**Weaknesses:**

They don’t consider generating the filling text that is not random (e.g. is about math and reasoning) to make the task more difficult.


They only focus on the tasks that have explicit assignment expressions in the text. As the LLMs get more advanced, we need the test cases where the information necessary for the reasoning is expressed in natural language and not explicitly specified with a special notation.

**Questions:**

How many tasks are included in this benchmark for each experiment?

Could you present the results from Table 1 in a way that clearly highlights which models are performing better?

I am curious about how the o1 model would perform compared to the others (and o1 mini).

Is the complexity of the problem only determined by the number of variables? How can you increase the complexity of the problem by changing the relationship of the variables and the nature of the problem (e.g. can some problems be unsolvable?)?

What would happen if the filler text were meaningful rather than randomly generated?

---

> ### Author Response · Authors · 2024-11-24
> **Official Response to ZEhP**
>
> Thank you so much for your attention and the time taken to read the paper. We appreciate your acknowledgment of the methodology and significance of the benchmark. We will try our best to address your concern on filler text and diversity of the benchmark, together with other questions. If you feel that your concerns have been accounted for, please consider increasing your score. For better readability, we rearrange the problem responses.
>
>  - Only symbolic assignment and limited diversity
>
> Thank you for such an insightful comment. To increase the diversity of our current dataset, we introduce a new problem-generation system for generating common-sense reasoning problems of large volumes with fine-grained control. (More details about the common sense problem generation in $\text{\textcolor{blue}{Section 4.2, page 6}}$) To summarize, we see the potential in mapping between computational graph and natural language problems from the old FACTOR and we apply its spirit onto real-world context. The result is the new FACTOR in the revised paper, containing three subsets of tasks classified based on hierarchical complexity in natural language structure (elaborated intensively in the paper). The easy subset contains the old FACTOR, while the medium and hard subsets contain the reasoning problem in the real-world context. The latter two contain problems similar to the ones in GSM-8K and have 19 splits of problems, ranging from #ops = 2 to #ops = 19). To further increase the diversity of the tasks, we use three different real-world context templates, together with both forward reasoning and backward reasoning modes. More details and evaluations can be read in the revised paper. Below, we show the results of Llama-3.1-70B-Instruct evaluation on these three subsets. You can discerningly see that the rate for the accuracy degradation is different, the Easy subset has the slowest decay rate, while the Hard subset decays the fastest.
>
> | op | Easy | Medium | Hard |
> |-----|--------|--------|--------|
> | 5 | 0.808 | 0.86 | 0.7 |
> | 8 | 0.832 | 0.875 | 0.512 |
> | 10 | 0.708 | 0.612 | 0.41 |
> | 12 | 0.628 | 0.56 | 0.36 |
> | 15 | 0.556 | 0.43 | 0.19 |
> | 20 | 0.516 | 0.2 | 0.05 |
>
>  - Filler text is random and not random is preferred
>
> The filter text is indeed vital for our benchmark. Together with our well-conceived noise, FACTOR medium and hard effectively prevent RAG from solving the question entirely. (Details about the noise addition in $\text{\textcolor{blue}{Section 4.4, page 7}}$). One sentence to summarize how our close noise is added: benefitting from having a problem generator in our design of the dataset construction. We carefully devise a method that uses the same problem generator to generate noise statements that is super close in semantics and format and have strong links to the essential logic statements that are supposed to be used to solve the problems.
>
> As a quantitative study of the noise quality, we conduct the following experiment. The experiment is based on FACTOR Medium, the commonsense reasoning subset. We compare two different types of noise: one is the aforementioned close noise generated by the problem generator, while another one is the noise I generated by asking the GPT-4o model to write a lengthy documentary related to the problem text. A RAG with 2K context with Llama 3.1 8B Instruct together with a full context Llama 3.1 8B Instruct on the 8K medium subset of FACTOR. Also, for the best theoretical performance of the RAG, we enable iterative prefilling of every token generated. The result is presented in the table below. As you can see, for long context LLM like Llama 3.1 8B Instruct, two types of noise behave similarly, the GPT-4O generated noise is even slightly harder in some ops. However, for the RAG, the close noise significantly worsens its performance compared to the generated noise, further proving the high quality of the noise.
>
> | Noise Type       | 8K Iterative Prefill RAG | GPT4o Documentary-like (RAG) | 8K (Llama-3.1-8B-Instruct) | GPT4o Documentary-like (Llama) |
> |-------------------|-----------|---------------|----------|------------|
> | 2| 0.52| 0.739 | 0.8024 | 0.717742|
> | 4| 0.22| 0.69 | 0.6794 | 0.512097 |
> | 5 | 0.0729 | 0.291| 0.2641 | 0.33871|
> | 6 | 0.02 | 0.24| 0.2338| 0.260081 |
> | 8  | 0.06| 0.26 | 0.2459 | 0.207661|
> | 10 | 0.03| 0.13 | 0.1370  | 0.157258 |
> | 12 | 0 | 0.07 | 0.0806 | 0.100806 |
>
>  - How many tasks and more details about the dataset construction
>
> For the old FACTOR (initially submitted), we only have one suite of tests that consists 39 different subsets of tests, they are from operation number of 1 to operation number of 39. After the expansion, the new FACTOR now consists of three suites: Easy, medium, and hard. While Easy is the old FACTOR. For medium and hard, each consists of 19 subsets from operation number of 2 to 19. For every data point presented, we test at least 200 examples of that kind.

---

> ### Author Response · Authors · 2024-11-24
> **Continue the response to ZEhP**
>
> - How many tasks are in FACTOR benchmark?
>
> For the old FACTOR (initially submitted), we only have one suite of tests that consists 39 different subsets of tests, they are from operation number of 1 to operation number of 39.
> After the expansion, the new FACTOR now consists of three suites: Easy, medium, and hard. While Easy is the old FACTOR. For medium and hard, each consists of 19 subsets from operation number of 2 to 19. For every data point presented, we test at least 200 examples of that kind.
>
>  - Could you present the results from Table 1 (now $\text{\textcolor{blue}{Table 5, page 9}}$) in a way that clearly highlights which models are performing better?
>
> We acknowledge that there is confusion in our dataset description. More can be found now in $\text{\textcolor{blue}{Section 5.1 page 9}}$ of the revised paper. We re-organize Table 1 in the newly edited paper. Previously, we had two metrics, which makes it less intuitive to compare between models. Now, we propose to compare using the area under the curve (AUC), while CDF and CDO are also presented for a better understanding of the model behavior.
>
> | Index | Model                     	| CDF 	| CDO 	| \( N_{\text{eff}} \) |   AUC  | AUC40 |  ELO |
> |-------|-------------------------------|---------|---------|----------------------|--------|-------|------|
> | 1 	| o1-mini                   	| -0.0117 | 0.6303  | 53.87            	| 139.34 | 38.25 | 1294 |
> | 2 	| Gemini-1.5-Pro-002        	| -0.0081 | -0.0696 | -8.59            	| 114.87 | 31.42 | ---  |
> | 3 	| GPT-4o-2024-05-13         	| -0.0220 | 0.2298  | 10.45            	| 55.90  | 31.31 | 1251 |
> | 4 	| GPT-4o-2024-08-06         	| -0.0205 | 0.0899  | 4.39             	| 53.17  | 29.45 | 1237 |
> | 5 	| Claude-3-5-Sonnet-20240620	| -0.0187 | -0.1447 | -7.74            	| 46.27  | 25.52 | 1268 |
> | 6 	| Qwen2.5-72B-Instruct      	| -0.0265 | 0.2056  | 7.76             	| 45.50  | 28.26 | 1223 |
> | 7 	| Mistral-Large-Instruct-2407   | -0.0279 | 0.2100  | 7.53             	| 43.37  | 28.12 | 1231 |
> | 8 	| Gemini-1.5-Flash-002      	| -0.0244 | -0.0180 | -0.74            	| 40.25  | 25.04 | ---  |
> | 9 	| GPT-4o-mini-2024-07-18    	| -0.0401 | 0.4303  | 10.73            	| 35.67  | 27.19 | 1219 |
> | 10	| GPT-4-Turbo-2024-04-09    	| -0.0378 | 0.2514  | 6.65             	| 33.10  | 25.06 | 1226 |
> | 11	| Llama-3.1-70B-Instruct    	| -0.0302 | -0.0481 | -1.59            	| 31.56  | 21.60 | 1187 |
> | 12	| Qwen2-72B-Instruct        	| -0.0467 | 0.0123  | 0.26             	| 21.67  | 17.91 | 1178 |
> | 13	| Claude-3-Haiku-20240307   	| -0.0471 | -0.0848 | -1.80            	| 19.50  | 16.87 | 1173 |
> | 14	| Mistral-Nemo-Instruct-2407	| -0.0608 | 0.0735  | 1.21             	| 17.66  | 15.85 | ---  |
> | 15	| Llama-3.1-8B-Instruct     	| -0.0694 | -0.4615 | -6.65            	| 9.08   | 8.10  | 1132 |
>
>  - o1?
>
> In our initial submission, we actually have an o1-mini result in $\text{\textcolor{blue}{Figure 3 (c), page 8}}$. You can see that normally, models would have a score near zero for operation nearly 40, while O1-mini can go past operation 160. Showing its strong reasoning ability.
> | op  | Accuracy |
> |-----|----------|
> | 10  | 1.000    |
> | 20  | 0.975    |
> | 30  | 0.975    |
> | 50  | 0.950    |
> | 60  | 0.775    |
> | 80  | 0.675    |
> | 100 | 0.600    |
> | 120 | 0.650    |
> | 140 | 0.300    |
> | 150 | 0.250    |

---

> > ### Author Response · Authors · 2024-11-27
> > **Response to ZEhP**
> >
> > We appreciate your attention to our work and your inspiring feedback! It is approaching the paper edition deadline. If you feel that any critical experiments or results are missing, please let us know immediately so we can try our best presenting them in the revised paper to dispel your concerns and clarify your confusions with additional soundness. Thank you so much.

---

### Official Review · Reviewer_wnwN · 2024-11-04

**Soundness:** 2
**Presentation:** 3
**Contribution:** 2
**Rating:** 3
**Confidence:** 3

**Summary:**

Authors introduce FACTOR, a benchmark to evaluate LLMs' long context complex reasoning ability. The goal was to distinguish between long-context LLMs and RAG and explore how LLMs reasoning abilities depend on context length and task complexity.

FACTOR manipulates two variables that control for task complexity and context length: the first one is the number of variables--the larger the number of variables the higher the complexity of the task; the second one is length of filler text--adding text to the necessary portion of logic arguments. The authors find log-linear relationship between accuracy and complexity. Filler text is randomly generated text.

Authors find that for some LLMs' performance is outlines by different performance zones. They also outline how different pertaining strategies can influence performance.

**Strengths:**

I like that authors introduce task complexity and analyse performance as a function of this. The authors also show a nice analysis of this performance and divide it into performance phases.

**Weaknesses:**

The main weakness of this paper is that it doesn't provide analysis or evidence for the claim it set out to explore at the outset: i.e. showing the 'unique reasoning capabilities of long-context LLMs unattainable by RAG methods'. They don't provide any analysis of LLMs using RAG to support this claim. Authors could've had a the models tested also use RAG to perform the same task. In particular, because their filler text is randomly generated, I suspect RAG wouldn't have issues finding the relevant text and which would then greatly simplify the task for an LLM.

Another smaller issues pertains to the interpretation of the context decay offset (i.e. CDO) which is the intersect of the fitted log performance line and captures 'the baseline performance level influenced by context length'. I don't think this would be a correct interpretation of the intercept, for a simple reason that these intercepts of log performance can be higher 0 (and many times are as seem from Table 1), meaning that that this baseline performance can be higher than 100%, which doesn't make sense. In the context of the paper, these intercepts have no interpretation in terms of performance.

**Questions:**

Why no experiments were done comparing LLMs that use long-context windows and those that use RAG?

Why was the filler text decided to be a completely random text? Why not some meaningful filler text?

---

> ### Author Response · Authors · 2024-11-24
> **Official Response to Reviewer wnwN**
>
> Thank you so much for your attention and time taken to read the paper and your insightful comments. We are very thankful that you like our paper’s initiative to evaluate model reasoning ability through fine-grained control of the level of difficulty. Also, your questions are very insightful in our development of the FACTOR benchmark. We revise our methods hoping to address your concern. If you feel that your concern has been taken into account, please consider raising your score. The question response orders are rearranged to better help the narrative and for better understanding.
>
>  - Filler text to be random text and RAG experiments missing?
>
> The filter text is indeed vital for our benchmark. Together with our well-conceived noise, FACTOR medium and hard effectively prevents RAG from solving the question entirely. (Details about the noise addition in $\text{\textcolor{blue}{Section 4.4, page 7}}$). One sentence to summarize how our close noise is added: benefitting from having a problem generator in our design of the dataset construction. We carefully devise a method that uses the same problem generator to generate noise statements that are super close in semantics and format and have strong links to the essential logic statements that are supposed to be used to solve the problems.
>
> | op  | RAG (with Llama 3.1 8B-Instruct) >2k | Llama-3.1-8B-Instruct 8K | RAG (with Llama 3.1 70B-Instruct) >2k | Llama-3.1-70B-Instruct 8K |
> |-----|--------------------------------------|---------------------------|---------------------------------------|---------------------------|
> | 2   | 0.13 | 0.802415                  | 0.175 | 0.975                     |
> | 3   | 0.11  | 0.532258                  | 0.07  | 0.835                     |
> | 4   | 0.055 | 0.679435                  | 0.145 | 0.92                      |
> | 5   | 0.095 | 0.264113                  | 0.1 | 0.87                      |
> | 6   | 0.08 | 0.233871                  | 0.12 | 0.705                     |
> | 7   | 0.065 | 0.280242                  | 0.085 | 0.655                     |
> | 8   | 0.055 | 0.245968                  | 0.05  | 0.59                      |
> | 9   | 0.05 | 0.169355                  | 0.07  | 0.38                      |
> | 10  | 0.045 | 0.137097                  | 0.045 | 0.39                      |
>
> The result of our high-quality noise addition is that RAG is no longer effective in solving our tasks. (Referring to revised Section 3 for more) We build a standard RAG system using Llama 3.1 8B Instruct as the decoder while using the al-mpnet-base-v2 for the retriever. We evaluated the system on various previous and standard long context datasets to show that the RAG delivers reasonable performance. Here we highlight the Variable Tracking (VT) from RULER, our RAG method gives 86% acc, whereas LLM directly gives 99%. We use problem context 8K while the budget for RAG is to be 2K. We tested this RAG system on our FACTOR medium subset, and we found that it quickly dropped to zero. We also replaced the decoder with Llama 3.1 70B Instruct, and we found that it is the same pattern. More details about our evaluation are in $\text{\textcolor{blue}{Section 3, page 3 - 5}}$.
>
> To make the case more convincing, we now enable iterative prefilling for the RAG side. We disregard the efficiency of the RAG system and test its performing ceiling. **It now reaches 100% on the VT**. However, it still struggles heavily on our FACTOR medium when op > 3. You can see that the accuracy quickly drops to zero. Therefore, RAG cannot determine the essential logic path in its chunk embedding space.
>
> | Metric               | 8K Iterative Prefill RAG | 8K (Llama-3.1-8B-Instruct) |
> |-----------------------|--------------------------|----------------------------|
> | VT from RULER        | 100%                    | 99.2%                     |
> | 2                    | 0.52                    | 0.8024                    |
> | 4                    | 0.22                    | 0.6794                    |
> | 5                    | 0.0729                  | 0.2641                    |
> | 6                    | 0.02                    | 0.2338                    |
> | 8                    | 0.06                    | 0.2459                    |
>
>  - CDO metric hard to understand?
> Thank you for raising your concern about the interpretation of the CDO. We realize that having the metric defined as the log-linear curve interception with the y-axis has the issue of interpretability. We shift and now use the measured score at op = 1 to be the value to replace CDO, measuring the baseline performance of a model in that context length. Having op = 1 essentially reduces the task to a retrieval task, removing the complexity of logic and effectively only measuring the model performance in the given context. We thank the reviewer for your insightful comments.

---

> > ### Author Response · Authors · 2024-11-27
> > **Response to wnwN**
> >
> > We appreciate your attention to our work and your inspiring feedback! It is approaching the paper edition deadline. If you feel that any critical experiments or results are missing, please let us know immediately so we can try our best presenting them in the revised paper to dispel your concerns and clarify your confusions with additional soundness. Thank you so much.

---

### Author Response · Authors · 2024-11-24
**Response to all reviewers**

We thank all reviewers [`R1 (wnwN)`, `R2 (ZEhP)`, `R3 (ALNr)`, `R4 (zsAb)`] for their attention and interest in reading FACTOR and for your constructive and supportive comments. We are glad to find that people think that this benchmark has strong significance `R3 (ALNr)`, and we are also pleased to see people appreciate our methodology to do fine-grained op control when benchmarking `R1 (wnwN)` and `R2 (ZEhP)`.

We want to assure the reviewers that we take their suggestions and criticism seriously. Therefore, we substantially rewrote the paper submitted based on the valuable feedback. The differences can be summarized in the following bullet points:

 - **Bring FACTOR’s Spirit to Common Sense Reasoning and Fusing General Language Context into the Problem Generation** [`R2 (ZEhP)`, `R3 (ALNr)`, `R4 (szAb)`] To increase the diversity of our current dataset, we introduce an entire common-sense reasoning benchmark suite. This suite follows the core spirit of FACTOR to generate text problems from curated computational graphs. To increase the diversity of the machine-generated dataset, we curated three different real-world scenarios as the context template and evaluated models to run on it with both forward and backward logic. The result is that we can use this problem generation system to generate infinite GSM-8K-like problems while also fine-grainedly controlling and easily scaling up the problem’s difficulty. Together with the previous symbolic benchmark suite, we now have the full machine-generated benchmark decoupling context and logic difficulty and easy to scale up in difficulty. Details about the implementation of graph generation and text attachment can be found in the revised paper $\text{\textcolor{blue}{Section 4.2 - 4.4, page 6 - 7}}$.

 - **Present Detailed RAG Experiments to Show that RAG Cannot Perform Well on Complex Long-Context Reasoning Task** [`R1(wnwN)` and `R2 (ZEhP)`]
We added extensive comparisons and studies to contrast RAG’s performance and long-context LLM on our tasks. We first build up a RAG system baseline using Llama 3.1 8B Instruct as the decoder. We test it across various popular and open-sourced long context benchmarks. For VT from RULER, the RAG achieves 86.4% accuracy with the full LLM achieving 99.2%. However, the RAG accuracy on our commonsense reasoning suite drops to zero when the number of operations (op) > 3. We further enable iterative prefilling of the RAG system so that it can achieve 100% on VT, but it still has near 0 accuracy when op > 3, despite the full LLM scoring >20% consistently. The same phenomenon holds true even when we use Llama 3.1 70B Instruct as the RAG system’s decoder, signifying our dataset has a high-quality design of noise. More details can be found in $\text{\textcolor{blue}{Section 3, page 3 - 5}}$ RAG vs Long Context Benchmarks.

 - **Refined Description of the Methods and Evaluation Metrics** [`R2 (ZEhP)`, `R3 (ALNr)`]
To present FACTOR in a more discerning and open way, besides presenting intensive details on the implementation of the dataset generator, we also elaborate on more details about noise addition design together with the evaluation prompt, template, and parsing in $\text{\textcolor{blue}{Section 4, page 5 - 8}}$. We believe that providing these details is conducive to a more in-depth evaluation of the significance of the dataset.

---

### Author Response · Authors · 2024-12-03
**A Call for AC's Discretion**

Dear Area Chair,

We genuinely appreciate all four reviewers' time and effort in reviewing and raising valuable suggestions about FACTOR. On the other hand, it is unfortunate and disheartening to see that despite our significant efforts and major paper revision to carefully and comprehensively clarify in our responses, only one further feedback is received during the entire rebuttal. We believe that the lack of engagement during the rebuttal made it unfair and extremely difficult for us to further communicate and bolster the value of our work. Therefore, we sincerely request AC's discretion on our work. To make it easier for AC's navigation through the long and detailed reviews and responses, here we provide a one-stop summary for all the major contributions of FACTOR and our responses to related concerns.

**Problem and FACTOR**

Despite a plethora of popular long-context benchmarks of LLM, we first show that these popular long-context tasks can be satisfactorily solved by a simple-to-build RAG system. Therefore, we believe that the tremendous cost of training the long-context LLM isn't justified by the existing benchmarks. To solve the problem, we developed FACTOR, a synthetic long-context benchmark that aims to evaluate the complex reasoning of LLMs with various context-length settings. Because of the problem curation and the noise design, the RAG system completely cannot solve the task, while LLM delivers a reasonable range of performance. We further study the LLM performance degradation with higher difficulty as well as with longer context length, and we report that a comprehensive list of LLMs we tested generally displayed a exponential decaying performance-to-difficulty relationship that can be neatly modeled by a fitted curve. This observation offers new insight into LLM's reasoning ability and its long-context ability.

**Main Concerns We Faced**

*Lack of diversity in the benchmark* (`ZEhP`, `ALNr`, `zsAB`)

In the initial version of FACTOR we submitted, the task is focused on variable assignments without any semantics in the problem context. In rebuttal, we incorporate real-world context in our newly added Medium and Hard subsets of FACTOR. The addition is based on a new observation: LLM performance naturally degrades when the problem involves more hierarchical relationships in the problem set. We now generate controlled commonsense reasoning problems that contain hidden operations through semantics. More interestingly, we show that the exponentially decaying trend observed on the simple variable assignment subset holds on the new commonsense subsets.

*Lack of RAG experiments and the use of not random noise* (`wnwN`, `ZEhP`, `ALNr`)

During the rebuttal, we generate noise directly by the same problem generator. Then, we present RAG experiments on our newly added medium and hard subsets with the new noise. We first test a conventional setup using an LLM as the decoder but with context retrieved by a sentence transformer. We show that even though under a generous setting of 8K context length with a 2K budget, the RAG performance is near zero for reasonable operation counts. Then, to make the case even more convincing, we enable iterative prefilling for every token. The RAG still gives near 0 performance, showing a significant gap to the full context LLM.

*Computational Expensive* (`zsAb`)

Admittedly, a sweep of all the operation counts from 0 to 20 (medium and hard) or 40 (easy) is costly. However, since FACTOR uses the fitted curve and its parameters to cross-compare different models, we show that only evaluating once every three or four operation numbers can fit the landscape nicely, reducing the total testing compute by 3-4x. Also, since we newly added the hard subsets where LLM performance degrades the fastest, measuring smaller number of operations would give complete landscape.

*The fitted curve is closer to logistic regression, not exponential decay* (`ALNr`)

We observe that there are two phases of the general trend of LLM performance degradation with respect to problem difficulty. First, it remains flat for easy problems, and as the difficulty increases, the performance drops sharply. Intuitively, using the logistic regression is a natural way to describe this relationship. However, through comprehensive experiments with high precision and costs, we show that the trend is closer to a step-wise function where the beginning is a constant, followed by an exponential decaying function across various models and FACTOR subsets.

-----
We hope the post provides a concise summary of the entire paper and rebuttal. With the advent of ChatGPT-O1 and similar models, the reasoning ability of LLM is becoming stronger. The community is in desperate need of a fine-grainedly controllable dataset that is easily scalable in both difficulty and context length. FACTOR is a pioneering work in this direction and would be filling the dire need of the community.

Sincerely,

Paper 7490 Authors

---

### Meta-Review · Area_Chair_1Whx · 2024-12-20

**Metareview:**

The paper introduces the FACTOR benchmark (Factoring Analysis of Complexity and Textual Context in Reasoning). Factor evaluates LLMs by independently varying task complexity and context length. The results are fitted with log-linear model, and three indicators are extracted from it. The reviews acknowledge that there is a significant effort in this paper, and the line of research is important. Unfortunately, two reviewers raised concerns about the contribution and the justification for some choices, and I fully agree. As the discussion between the authors and the reviewers indicates, this needs to be clarified and will be a major rewrite, as a lot of changes and new ideas and arguments have been presented in the discussion. Moreover, the evaluation should involve some significant analysis of the log-linear model fits as well as of the predictive performance. Please note that the overall judgment should not be taken as a statement regarding the usefulness of your research.

**Additional Comments On Reviewer Discussion:**

One discussion arose from issues raised in the reviews. Unfortunately, it did not change the mind of the reviewer. The overall decision takes the reviews and rebuttal into account.

---

### Decision · Program_Chairs · 2025-01-22

Reject